# Sub-millisecond conformational dynamics of the A$_{2A}$ adenosine receptor revealed by single-molecule FRET

Ivan Maslov [1,2,3], Oleksandr Volkov [13], Polina Khorn [1], Philipp Orekhov [4], Anastasiia Gusach[1,11], Pavel Kuzmichev[1], Andrey Gerasimov[1,5], Aleksandra Luginina [1], Quinten Coucke [3], Andrey Bogorodskiy[1], Valentin Gordeliy[6], Simon Wanninger [7], Anders Barth [7,12], Alexey Mishin[1], Johan Hofkens[3,8], Vadim Cherezov [9], Thomas Gensch[3], Jelle Hendrix [2,3✉] & Valentin Borshchevskiy [1,10✉]

The complex pharmacology of G-protein-coupled receptors (GPCRs) is defined by their multi-state conformational dynamics. Single-molecule Förster Resonance Energy Transfer (smFRET) is well suited to quantify dynamics for individual protein molecules; however, its application to GPCRs is challenging. Therefore, smFRET has been limited to studies of inter-receptor interactions in cellular membranes and receptors in detergent environments. Here, we performed smFRET experiments on functionally active human A$_{2A}$ adenosine receptor (A$_{2A}$AR) molecules embedded in freely diffusing lipid nanodiscs to study their intramolecular conformational dynamics. We propose a dynamic model of A$_{2A}$AR activation that involves a slow (>2 ms) exchange between the active-like and inactive-like conformations in both apo and antagonist-bound A$_{2A}$AR, explaining the receptor's constitutive activity. For the agonist-bound A$_{2A}$AR, we detected faster (390 ± 80 μs) ligand efficacy-dependent dynamics. Our work establishes a general smFRET platform for GPCR investigations that can potentially be used for drug screening and/or mechanism-of-action studies.

[1] Research Center for Molecular Mechanisms of Aging and Age-Related Diseases, Moscow Institute of Physics and Technology, Dolgoprudny Moscow Region, Russia. [2] Dynamic Bioimaging Lab, Advanced Optical Microscopy Centre, Biomedical Research Institute, Agoralaan C (BIOMED), Hasselt University, Diepenbeek, Belgium. [3] Laboratory for Photochemistry and Spectroscopy, Division for Molecular Imaging and Photonics, Department of Chemistry, KU Leuven Leuven, Belgium. [4] Faculty of Biology, Shenzhen MSU-BIT University, Shenzhen, China. [5] Vyatka State University, Kirov, Russia. [6] Institut de Biologie Structurale J.-P. Ebel, Université Grenoble Alpes–CEA–CNRS, Grenoble, France. [7] Physical Chemistry, Department of Chemistry, Center for Nano Science (CENS), Center for Integrated Protein Science (CIPSM) and Nanosystems Initiative München (NIM), Ludwig-Maximilians-Universität Munich, Munich, Germany. [8] Max Plank Institute for Polymer Research, Mainz, Germany. [9] Bridge Institute, Department of Chemistry, University of Southern California, Los Angeles, CA, USA. [10] Joint Institute for Nuclear Research, Dubna, Russian Federation. [11] Present address: MRC Laboratory of Molecular Biology, Cambridge, UK. [12] Present address: Department of Bionanoscience, Kavli Institute of Nanoscience, Delft University of Technology, HZ Delft, The Netherlands. [13] Unaffiliated. ✉email: jelle.hendrix@uhasselt.be; borshchevskiy.vi@phystech.edu

G-protein-coupled receptors (GPCRs) constitute the largest superfamily of membrane proteins in humans containing over 800 members, which mediate critical physiological processes, such as neurotransmission, homeostasis, inflammation, reproduction, olfaction, vision, taste, and others[1,2]. GPCRs recognize a large variety of endogenous extracellular signaling molecules transmitting their corresponding signals inside the cell, and this process can be modulated by synthetic ligands or drug molecules. In fact, over 30% of all FDA-approved drugs target GPCRs[3]. Multiple lines of evidence suggest that the molecular mechanism of GPCR activation extends beyond a simple "on/off" mode. First, apo receptors show basal activity that can be suppressed by inverse agonists[4]. Second, different agonists vary in efficacy and can stimulate receptor activity to a different extent[5]. Third, a single receptor can signal through several intracellular pathways, some of which could be preferentially activated by so-called "biased" ligands[6]. These three phenomena indicate that receptors are highly dynamic molecules and sample several active and inactive states stochastically (for review, see refs. [7–9]).

The $A_{2A}$ adenosine receptor ($A_{2A}AR$) is expressed in many organs and tissues including those in the immune system, basal ganglia, heart, lungs, and blood vessels[10]. Throughout the body, $A_{2A}AR$ regulates the cardiovascular tonus causing vasodilation and promotes healing of inflammation-induced injuries by suppressing immune cells[11,12]. In the brain, $A_{2A}AR$ modulates dopamine and glutamate neurotransmission[12]. $A_{2A}AR$ is a promising target for drugs against insomnia, chronic pain, depression, Parkinson's disease, and cancer[12,13]. On the molecular level, $A_{2A}AR$ is activated by the endogenous extracellular agonist adenosine and initiates the cAMP-dependent signaling pathway via $G_s$ and $G_{olf}$ proteins[12,14]. Besides G proteins, $A_{2A}AR$ interacts with numerous other partners including GRK-2 kinase, β-arrestin, and other GPCRs[14,15]. One cryoEM and over 50 high-resolution X-ray crystallographic structures are available for antagonist- or agonist-bound $A_{2A}AR$ and for its ternary complex with an agonist and an engineered G protein, making this receptor an excellent model system for investigating GPCR structural dynamics. While static structures provide critical information about the receptor's lowest energy states, our understanding of the $A_{2A}AR$ function remains critically incomplete without detailed knowledge of its conformational dynamics.

The current information about $A_{2A}AR$ conformational dynamics is based mostly on several reported NMR experiments[16–24]. In response to ligand binding, different $A_{2A}AR$ amino acids either alter their sole stable conformations or vary relative probabilities of coexisting stable conformations[16,17]. On the picosecond-to-nanosecond timescale, some $A_{2A}AR$ amino acids increase side-chain dynamics, while others become stabilized[18]. Sub-millisecond conformational variability was shown for both apo-form[19] and agonist-bound $A_{2A}AR$[16,17,20]. Large-scale conformational changes in $A_{2A}AR$ with dwell times of seconds were also reported[19,21], but two independent studies described the corresponding long-lived states differently: in one report[19], a 3-state model with an attributed basal activity of 70% was proposed, while in the other[21], the authors put forward a 4-state model with a negligible basal activity. Thus, the current picture of $A_{2A}AR$ dynamics is complex and contradictory.

Studies of $A_{2A}AR$ dynamics face two major challenges: first, the need to cover a wide range of timescales from nanoseconds to seconds, and next, the difficulty to untangle multiple protein states within the ensemble. Single-molecule fluorescence spectroscopy provides tools to address both of these difficulties. Depending on the applied method, the fluorescence signal from individual receptors can be tracked with as low as a nanosecond temporal resolution for a total duration of either millisecond in case of freely diffusing molecules or even seconds to minutes using immobilized molecules[7,25].

Single-molecule fluorescence spectroscopy methods have been previously applied to study GPCR conformational dynamics[7]. For example, environmentally sensitive fluorescent dyes have been used as single-molecule reporters of conformational changes in the $β_2$ adrenergic receptor ($β_2AR$)[26–29], visual rhodopsin[30,31], and, more recently, $A_{2A}AR$[32]. Single-label experiments are attractive because of a minimal influence of the dye on the native receptor dynamics, but the experimental readouts are often limited and lack detailed structural interpretation. In addition, the results of single-label experiments can be obscured by multi-state dye photophysics. Another approach, based on single-molecule Förster Resonance Energy Transfer (smFRET) between two dyes can provide more direct structural outcomes and introduce additional internal controls, however, at the expense of double-labeling. smFRET has been shown to be especially useful to investigate structural dynamics of GPCR dimers[33–37]. To our knowledge, at the moment of this writing, ref. [38] is the only published application of smFRET to quantifying intramolecular conformational dynamics in GPCRs; this study addressed structural changes on the intracellular side of immobilized $β_2AR$ in detergent micelles.

Here, we applied smFRET to investigate the conformational dynamics of $A_{2A}AR$ in lipid nanodiscs freely diffusing in solution without immobilization. Using the MFD-PIE (multiparameter fluorescence detection with pulsed-interleaved excitation) technique[39] (Fig. 1a), we tracked the relative movements of two dyes attached to the intracellular tip of the transmembrane helix TM6 (L225C[6.27], superscripts indicate Ballesteros–Weinstein numbering[40]) and to the C-terminal intracellular helix H8 (Q310C[8.65]) of $A_{2A}AR$ (Fig. 1b). We observed that FRET efficiency in the double-labeled $A_{2A}AR$ increases upon agonist binding (Fig. 1c). Several burst-wise fluorescence analysis approaches—plot of burst-wise FRET efficiency against donor fluorescence lifetime[41], FRET 2-Channel kernel-based Density Estimator (FRET-2CDE)[42], Burst Variance Analysis (BVA)[43], and filtered Fluorescence Correlation Spectroscopy (fFCS)[44]—subsequently revealed sub-millisecond conformational dynamics of $A_{2A}AR$. Based on quantitative analysis of the obtained data for the receptor in its apo-state and upon addition of the inverse agonist ZM241385, the partial agonist LUF5834, or the full agonist NECA to the receptor, we finally propose a dynamic model of $A_{2A}AR$ activation.

## Results

**Labeling and reconstitution of $A_{2A}AR$ in nanodiscs.** To track the conformational dynamics of $A_{2A}AR$ with smFRET we chose to attach two fluorescent dyes to mutated residues L225C[6.27] on the intracellular end of TM6 and Q310C[8.65] on the C-terminal end of H8 (Fig. 1b and Supplementary Fig. 1a). In previous $A_{2A}AR$ FRET studies, a fluorescent protein-based FRET donor and fluorescent molecule based acceptor in similar labeling positions were shown to provide sufficient contrast between the active and inactive receptor states in live cells[45,46]. The L225[6.27] position is also homologous to the native cysteine C265[6.27] in $β_2AR$ that has been frequently used for fluorescent labeling[26–28,47–51].

We expressed the double-Cys mutant (L225C[6.27]/Q310C[8.65]) of $A_{2A}AR$ in *Leishmania tarentolae* and simultaneously labeled it with two maleimide-functionalized dyes, Alexa488 and Atto643 ("Protein expression, purification and labeling" in "Methods"). The wild-type (WT) $A_{2A}AR$ has six unpaired cysteines in its transmembrane helices (Supplementary Fig. 1a). To achieve specific labeling of the two genetically introduced cysteines, but spare the transmembrane native cysteines, we labeled the receptors in isolated cell membranes, as described previously[52].

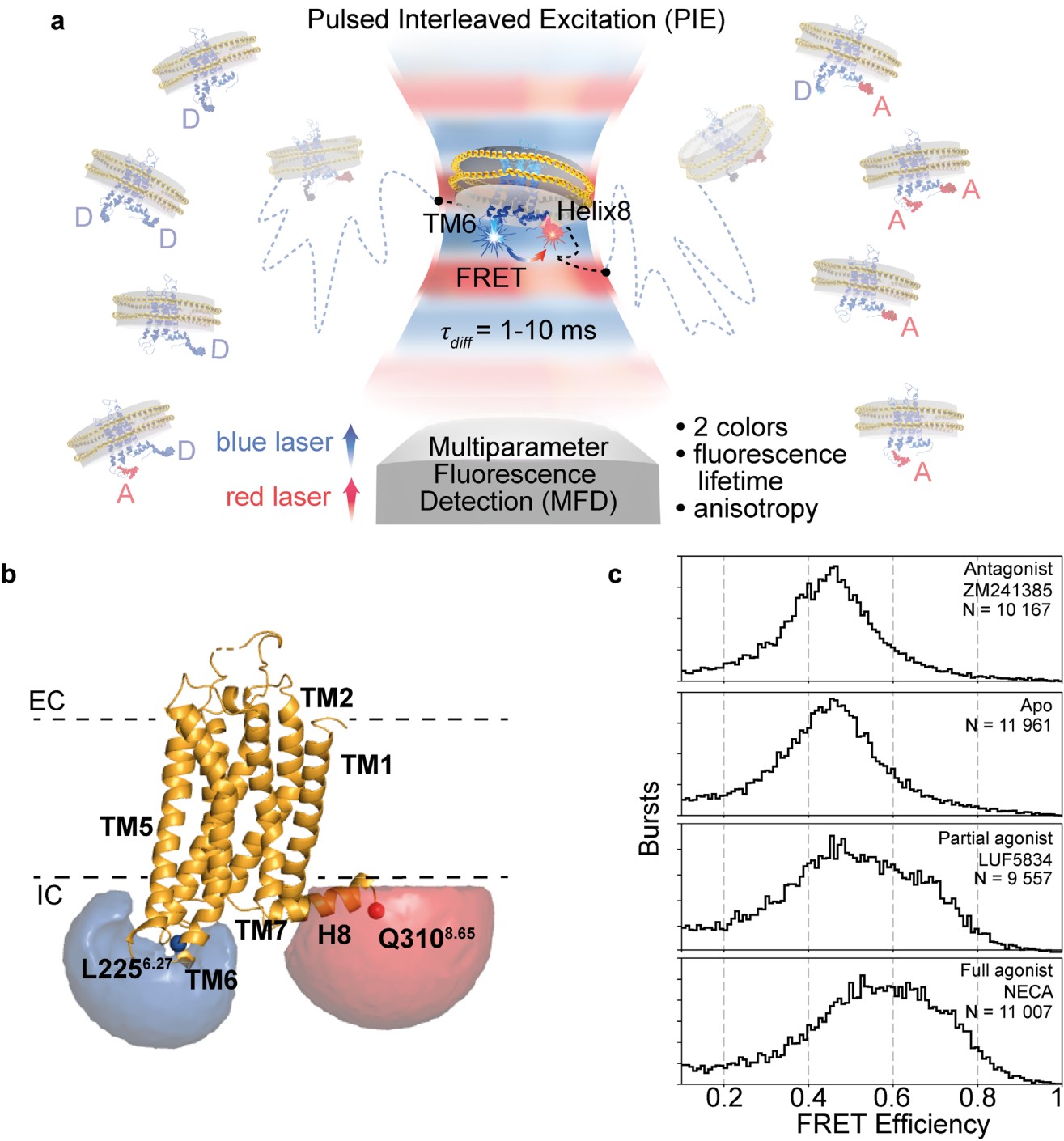

**Fig. 1 Agonist-induced conformational changes in A$_{2A}$AR are revealed by smFRET. a** Schematic illustration of the MFD-PIE smFRET experiment on A$_{2A}$AR embedded in lipid nanodiscs and stochastically labeled with the donor (Alexa488) and the acceptor (Atto643) fluorescent dyes at TM6 and H8. Eight coexisting labeling variants of A$_{2A}$AR are shown as shadowed receptors in both sides of the image, "D" and "A" correspond to donor and acceptor dyes, respectively. A$_{2A}$ARs diffuse in solution and stochastically cross the focal spot of an inverted fluorescence microscope. Bursts of fluorescence from donor and acceptor fluorophores are recorded within the 1–10 ms residence time of individual A$_{2A}$ARs crossing the focal spot. Only those receptors labeled with both, donor and acceptor, produce FRET signal. In the PIE approach, two spatially overlapped and alternatively pulsing lasers are focused by the microscope objective to excite donor and acceptor fluorescence consecutively. Using the MFD approach, fluorescence signals of the donor and acceptor are recorded separately, and the fluorescence lifetime and anisotropy of each dye are determined. **b** The labeled sites (L225$^{6.27}$, Q310$^{8.65}$) and the volume accessible for the dyes (simulated using FPS software[106]) are shown on the A$_{2A}$AR structure (PDB: 3EML[69]), the extracellular (EC) and intracellular (IC) membrane boundaries are obtained from the PPM web server[107] and shown as dashed lines. **c** Burst-wise distributions show an agonist-induced increase in FRET efficiency in the double-labeled A$_{2A}$AR. The number of bursts used for the analysis (N) is given for each condition.

After labeling, the receptors were purified and reconstituted in MSP1D1 nanodiscs, which can accommodate only a single monomeric receptor per nanodisc ("Nanodisc reconstitution" in "Methods")[53].

Size-exclusion chromatography confirmed a high purity and monodispersity of the nanodisc-reconstituted $A_{2A}AR$ samples (Supplementary Fig. 1b). Labeling specificity was confirmed with the WT receptor, which showed only a marginal dye fluorescence associated with the protein after the labeling procedure (Supplementary Fig. 1b and Supplementary Table 1). In both ensemble spectra and lifetime measurements of the fluorescently labeled $A_{2A}AR$ FRET-sensitized acceptor emission was readily observed, proving the existence of double-labeled FRET-active molecules in the samples (Supplementary Fig. 1c, d, "Fluorescence spectra characterization" and "Ensemble-based fluorescence lifetime measurements" in "Methods").

To test whether the double-cysteine mutant $A_{2A}AR$ (L225C[6.27]/Q310C[8.65]) is functional, we measured the ligand-induced thermostabilization of the isolated receptors as well as the agonist-induced cAMP accumulation in living cells. A fluorescent thermal stability assay[54] showed that the addition of either the antagonist ZM241385 or the agonist NECA in saturating concentrations increased the melting temperature of both WT and mutant $A_{2A}AR$ with respect to the apo-state by >7 °C, indicating ligand-binding activity of the receptor (Supplementary Fig. 1e, "Thermal shift assay" in "Methods"). A BRET assay of cAMP accumulation in HEK293T cells transiently expressing $A_{2A}AR$ showed very similar $pEC_{50}$ values (mean ± SD, three biological replicas) for both WT ($6.41 \pm 0.15$) and double-mutant ($6.45 \pm 0.06$) forms of the receptor upon stimulation with the agonist NECA (Supplementary Fig. 1f, "Measurement of $A_{2A}AR$ surface expression and Gs-signaling" in "Methods"). Although several previous studies reported an order of magnitude higher potency of NECA against WT $A_{2A}AR$ in CHO cells[55–57], $pEC_{50}$ values similar to those obtained here were measured in yeasts[58] and in membrane pellets isolated from CHO cells[59]. The mutant form of $A_{2A}AR$ retained ligand-binding activity in nanodiscs and signaling activity in HEK293T cells, therefore we assume that the conformational dynamics observed for the double-labeled receptor in smFRET experiments represent the native dynamics of the WT receptor.

**smFRET reveals ligand-induced conformational changes in $A_{2A}AR$.** We diluted fluorescently labeled $A_{2A}AR$ to nanomolar concentrations, mounted the sample on a microscope cover slip and recorded fluorescence intensity, lifetime, and anisotropy data from individual molecules diffusing freely across the femtoliter-sized observation spot (approximated by a 3D Gaussian with half-widths 0.5 μm, 0.5 μm and 2 μm) of a confocal fluorescence microscope (Fig. 1a, "Confocal MFD-PIE setup" and "smFRET data recording" in "Methods"). Inside the spot, donor and acceptor fluorophores are excited alternatingly using a two-color pulsed-interleaved excitation (PIE)[60]. The residence time of individual molecules (~1–10 ms) in the laser spot sets the upper limit of timescales approachable for the observation of $A_{2A}AR$ conformational dynamics. Using a 4-detector MFD scheme (Supplementary Fig. 2), photons detected from individual molecules were digitally tagged with (1) the spectral band in which they were detected, (2) their global arrival time with microsecond accuracy, (3) their relative arrival time with respect to the laser pulses within a ps-ns range, and (4) their optical polarization[61]. PIE, together with two-color detection, allowed us to distinguish double-labeled receptors (simultaneously labeled with donor and acceptor) from "donor-only" and "acceptor-only" receptors (Supplementary Fig. 3, "Burst identification" and" Selection of

double-labeled, donor-only and acceptor-only subpopulations" in "Methods").

The fraction of $A_{2A}ARs$ simultaneously labeled with donor and acceptor fluorophores showed different distributions of FRET efficiency depending on the bound ligand (Fig. 1c, "FRET efficiency and Stoichiometry" and "Correction factors" in "Methods"). The antagonist ZM241385 did not change FRET efficiency distribution within experimental error. On the contrary, both the partial agonist LUF5834 and the full agonist NECA shifted the mean FRET efficiency to larger values and increased the overall distribution width, compared to the apo-receptor. The increase in FRET efficiency was less pronounced for the partial agonist LUF5834 than for the full agonist NECA.

**Fluorescence lifetime data suggest sub-millisecond conformational dynamics of $A_{2A}AR$.** Besides fluorescence intensity, FRET is also reflected in fluorescence lifetime data. A two-dimensional plot of the per-burst FRET efficiency against the donor fluorescence lifetime provided the insights into the receptor's conformational dynamics (Fig. 2a, b, "Burst-wise fluorescence lifetime" in "Methods"). In theory, data for rigid molecules, in which FRET efficiency remains constant over the duration of a burst should be distributed along a curved diagonal line that intersects the lifetime axis at the lifetime of the donor-only population and the FRET efficiency axis at unity, commonly referred to as the "static FRET line" (Fig. 2a). Alternatively, if receptor molecules sample different conformations during their residence time in the focal spot (1–10 ms) on a timescale that is longer than the nanosecond fluorescence lifetime, their bursts should be shifted from the "static FRET line" toward the longer lifetime region. This phenomenon can be explained by the higher weights of the lower FRET states in the fluorescence lifetime averaging due to the larger number of photons emitted by the donor. The observed rightward deviations of our burst data from the static FRET line are statistically significant and indicate the existence of sub-millisecond conformational dynamics (beyond the fast dynamics expected for dye linkers) in the apo as well as agonist- and antagonist-bound states of $A_{2A}AR$ (Fig. 2b and Supplementary Fig. 4).

**FRET-2CDE and BVA confirm that agonists enhance conformational dynamics in $A_{2A}AR$ compared to apo-receptor.** Variations of FRET efficiency within fluorescence bursts from individual receptors suggest the presence of conformational dynamics. To analyze these variations further we used two complementary approaches: FRET-2CDE[42] and BVA[43]. Both methods assign dynamics scores to individual molecules and are sensitive to the dynamics that are slower than the time used for FRET efficiency averaging (roughly 100 μs for both approaches).

The FRET-2CDE score provides an unbiased way for the separation of static and dynamic subpopulations of molecules and for the comparison of their fractions in different datasets[42] ("FRET-2CDE analysis" in "Methods"). The main advantage of FRET-2CDE is that it is minimally influenced by the mean FRET efficiency in a dynamic molecule. Theoretically, static molecules should have FRET-2CDE ≈ 10, while higher FRET-2CDE values correspond to more pronounced conformational dynamics (Fig. 2c). In our data, neither the apo nor ligand-bound $A_{2A}AR$ showed a clear separation of different receptor subpopulations along the FRET-2CDE axis, but the observed deviations of FRET-2CDE scores from those expected for fully static molecules were statistically significant (Supplementary Fig. 5). In addition, agonists increased the mean FRET-2CDE score of $A_{2A}AR$ compared to the apo or antagonist-bound receptors (Supplementary Table 2). The fraction of receptors that exceeded the

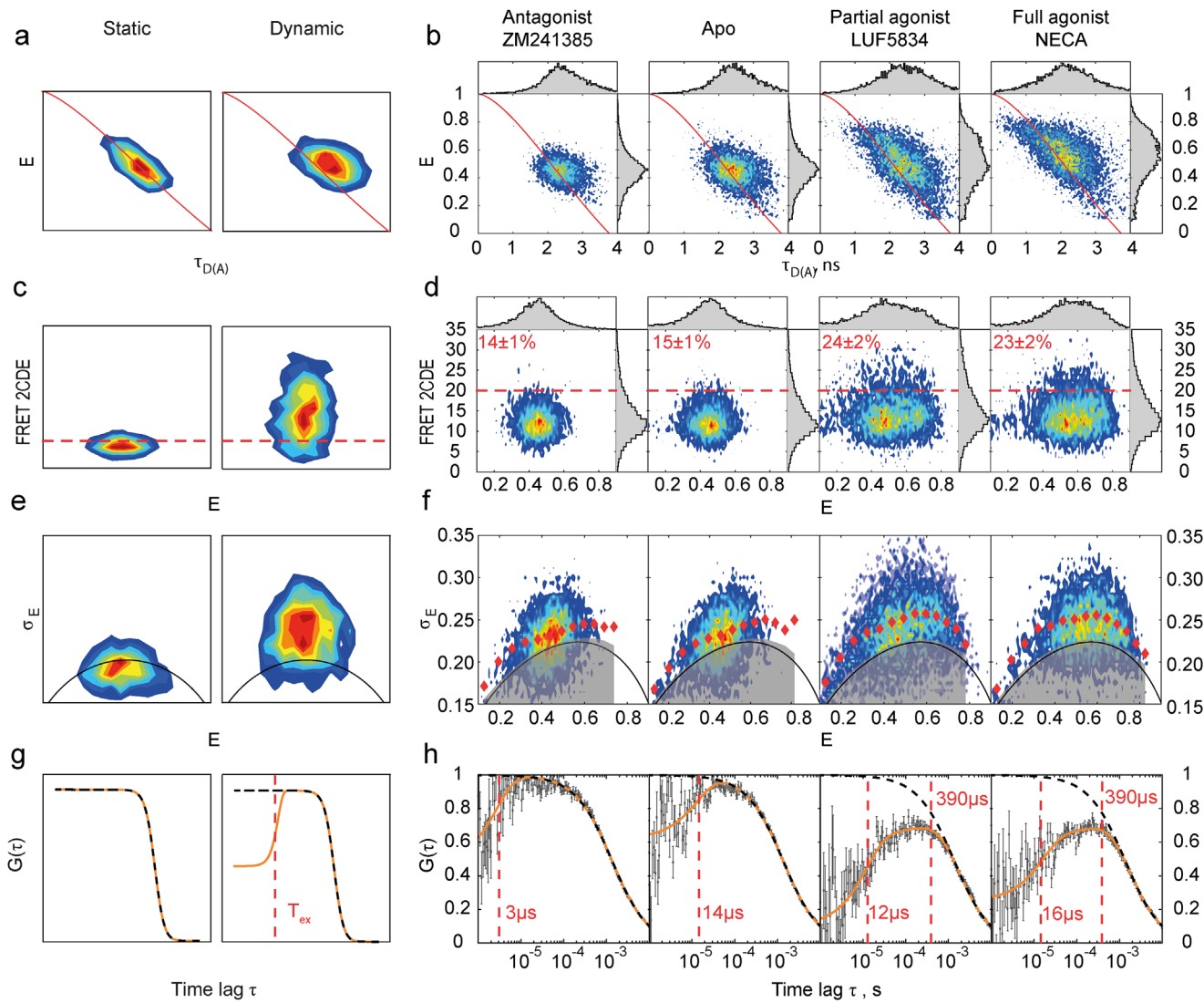

**Fig. 2 Four complementary burst-wise analysis approaches suggest an agonist-induced increase in the sub-millisecond conformational dynamics of A$_{2A}$AR.** Contour plots are two-dimensional histograms of different fluorescence burst parameter distributions. The qualitative differences between "static" and "dynamic" molecules expected in each analysis approach are shown in the drawings (**a**, **c**, **e**, **g**). The experimental data for double-labeled A$_{2A}$AR are shown in the plots (**b**, **d**, **f**, **h**). **a**, **b** The FRET efficiency is plotted against donor fluorescence lifetime. The 'static FRET' line is shown in red. A shift of burst distribution to the right from the red line indicates dynamic FRET. **c**, **d** The FRET-2CDE dynamics score is plotted against FRET efficiency $E$. The FRET-2CDE = 20 threshold is indicated as red dashed lines, and the percentage of bursts with FRET-2CDE > 20 is shown in red text (mean ± SD, three technical replicas with different protein aliquots). **e**, **f** BVA dynamics scores are plotted against FRET efficiency. Red diamonds show the centers of burst subgroups equally spaced along the FRET efficiency axis. The solid black lines show mean BVA scores, and the transparent gray areas demonstrate 99.9% confidence intervals expected for static molecules, given the shot noise present in the data. **g**, **h** The cross-correlation fFCS function is plotted against time lag. Experimental points with error bars are shown in gray; the error bars are SDs obtained after splitting the photon data into ten equally sized bins and correlating each individually. The fitting curves are shown in orange; the diffusion-related terms are shown as dashed black lines; the exchange times derived from the fit are highlighted with vertical red lines. $\chi_{red}^2$ of the global fit is 1.1. The source data is available online as Supplementary Data 1. The number of fluorescence bursts used for the analysis are the same as for Fig. 1c: 10,167 for ZM241385, 11,961 for apo-state, 9557 for LUF5834, 11,007 for NECA.

threshold FRET-2CDE > 20 was also higher for the agonist-bound receptors: 14 ± 1% for ZM241385, 15 ± 1% for apo, 24 ± 2% for LUF5834, 23 ± 2% for NECA (Fig. 2d). The fraction of molecules with high-FRET-2CDE scores is larger for the agonist-bound receptor for any threshold value (Supplementary Fig. 6). These results indicate that either the amplitude of the observed dynamics or the number of the inter-state transitions per burst increase in A$_{2A}$AR upon agonist binding.

BVA provides a statistically robust way to test whether the observed variations of FRET efficiency exceed fluctuations expected from the shot noise and thus to prove conformational

dynamics[43]. To apply BVA, we split bursts into consecutive photon windows with $n = 5$ photons in each (~100 µs long), calculated standard deviations of the bin-wise FRET efficiencies within each burst, and plotted them against the mean FRET efficiency ("Burst Variance Analysis (BVA)" in "Methods"). The obtained BVA scores exceeded the 99.9% confidence interval expected from the shot noise under all four conditions (Fig. 2e, f), therefore, BVA confirmed that sub-millisecond conformational dynamics are already present in the apo and antagonist-bound A$_{2A}$AR and further increased in the agonist-bound A$_{2A}$AR.

**fFCS reveals fast photophysics-related dynamics and slow agonist-induced dynamics in $A_{2A}AR$.** To estimate the timescales of $A_{2A}AR$ conformational dynamics we applied the fFCS approach[44] ("Filtered Fluorescence Correlation Spectroscopy (fFCS)" in "Methods"). We used the photon arrival time and anisotropy information to split the photon stream from the double-labeled molecules in silico between the low-FRET (LF) and high-FRET (HF) channels (Supplementary Fig. 7). Theoretically, if the LF and HF species are just two extremes of a heterogeneous ensemble of long-lived receptor states, then cross-correlation between the two channels will show only a diffusion-based sigmoidal component decreasing with the correlation lag time (Fig. 2g). Contrarily, if the LF and HF species interconvert on the µs–ms timescales, the cross-correlation should be lower in the time lag region shorter than the state exchange time (Fig. 2g). For all four conditions, fFCS curves deviate from the diffusion-related sigmoidal trend (Fig. 2h, Supplementary Data 1). For the apo and antagonist-bound receptor, the deviations (later called anticorrelations) are pronounced only in the 1–100 µs timescale, while for the agonist-bound protein the anticorrelation is also apparent in the 100–1000 µs timescale.

In the 1–100 µs timescale, fFCS-anticorrelation is expected from donor and acceptor photoblinking, as was also evident in sub-ensemble FCS analyses of single-labeled molecules for all apo/ligand-bound conditions (Supplementary Fig. 8). Dynamics of the dye linkers and local fluctuations of protein structure may also contribute to anticorrelation at this fast timescale. In the 100–1000 µs timescale, no dynamics were detected in sub-ensemble FCS for single-labeled molecules (Supplementary Fig. 8). This proves that the observed agonist-induced FRET dynamics are not due to dye photophysics, but must arise from sub-millisecond protein dynamics.

To quantify the exchange time of these dynamics, we fitted the anticorrelation terms in the fFCS curves with exponential decays. Initially, we employed just one anticorrelation term for each dataset, and optimized the diffusion time $\tau_{diff}$ globally across all four datasets. This fit adequately described the data for the apo/antagonist-bound $A_{2A}AR$, but showed systematic deviations in the 1–10 µs region for the agonist-bound $A_{2A}AR$ (Supplementary Fig. 9, Supplementary Table 3A, $\chi_{red}^2 = 1.5$). For this reason, we introduced a second anticorrelation term in the fitting model for the agonist-bound $A_{2A}AR$.

In this way, with one anticorrelating term for the apo and antagonist-bound $A_{2A}AR$ and two anticorrelating terms for the agonist-bound receptor, we obtained a satisfactory fit (Fig. 2h and Supplementary Table 3B, $\chi_{red}^2 = 1.1$). The fast anticorrelaction term ($A_1$) was present in all four datasets and was assigned mostly to dye photophysics; the slow anticorrelation term ($A_2$) appeared only in the agonist-bound receptor data. This fFCS model adequately describes the experimental data for all conditions and provides the exchange time of slow agonist-induced dynamics $\tau_2 = 390 \pm 80$ µs (error was estimated as a half-width of the 95% confidence interval of the fitting).

**PDA quantifies populations of active-like and inactive-like states in dynamic $A_{2A}AR$.** To quantify the populations of $A_{2A}AR$ in different FRET states in the apo and ligand-bound forms we used the photon distribution analysis (PDA) method[62,63] ("Photon distribution analysis (PDA)" in "Methods"). In contrast to multi-state Gaussian fitting, PDA explicitly describes FRET data by taking into account the background, shot noise and receptor dynamics. For PDA, we split the fluorescence bursts into time bins of constant duration (0.5 ms, 1 ms, and 2 ms) and analyzed them globally across apo and ligand-bound conditions. In dynamic systems, a molecule can sample several states during an individual time bin, and therefore, the FRET efficiency distribution depends on the duration of the time bin. PDA is most sensitive for picking up interconversion times on the diffusion timescale (1–10 ms); for faster or slower dynamics, PDA can, however, still be constrained a priori to demonstrate that the proposed model of the conformational space does not contradict the observed FRET efficiency distributions. All models with less than three states produced a poor fit of experimental data with $\chi_{red}^2 > 10$. The best among them was a model with two interconvertible states providing $\chi_{red}^2 = 10.3$ (Supplementary Fig. 10, Supplementary Table 4). Meanwhile, a three-state PDA model with three static states described the experimental distributions well ($\chi_{red}^2 = 3.2$, Supplementary Fig. 11 and Supplementary Table 5).

Despite the low $\chi_{red}^2$, the static three-state PDA model contradicts our findings from fFCS analyses, where fast dynamics were present in all apo/ligand-bound receptor forms and additional slow dynamics appeared in the agonist-bound $A_{2A}AR$. We, therefore, subsequently examined whether an fFCS-inspired model could equally well describe the experimental data in PDA. Since PDA is insensitive to fast (<20 µs) dynamics observed in fFCS, the fitting model for the apo and antagonist-bound $A_{2A}AR$ included only three static states. For the agonist-bound $A_{2A}AR$, we introduced a slow dynamics component with a fixed exchange time ($\tau_2 = 390 \pm 80$ µs, as observed in fFCS) between two states, while keeping the third one static. This fFCS-constrained PDA model adequately described the experimental data ($\chi_{red}^2 = 3.6$, Fig. 3a, Supplementary Fig. 12, Supplementary Table 6, and Supplementary Data 2). The difference between $\chi_{red}^2$ for the fully static and fFCS-constrained dynamic models (3.2 and 3.6, respectively) was insignificant within experimental error ("Photon distribution analysis (PDA)" in "Methods"). Together, both fFCS and PDA can thus be consistently described by the same unified kinetic model. We further used this model to quantify the populations of the $A_{2A}AR$ states.

Using the fFCS-constrained model in PDA we determined the mean values and variances of the FRET efficiency and populations for each PDA state under the apo and ligand-bound conditions (Fig. 3a, Supplementary Fig. 12, and Supplementary Table 6). PDA converged to a model, where the static state exhibited the lowest FRET efficiency (LF) and the interconvertible states (MF and HF) possessed a medium and high-FRET efficiency, respectively. The PDA results (Fig. 3a) revealed that both agonists increased the population of the highest FRET efficiency state (HF state), and decreased the population of the state with intermediate FRET efficiency (MF) state—therefore, we assume that the HF and MF states correspond to the active-like and inactive-like conformations of $A_{2A}AR$, respectively. Approximately 10–20% of $A_{2A}AR$ molecules always stay in the low-FRET (LF) state independently of the added ligand—we assign this fraction to receptors locked in a long-lived non-functional state or improperly folded protein.

Interestingly, the active-like HF state is also observed in the apo-receptor ensemble and even in the ZM241385-bound receptors. Additionally, the sample with the full agonist NECA has a higher population of the active-like HF state compared to the partial agonist LUF5834. The small variations in state populations between the apo-receptor and the antagonist-bound receptor are below statistical significance. We discuss below the implications of these results on the basal activity, partial agonism, and inverse agonism in $A_{2A}AR$.

## Discussion

In this study, we used smFRET to investigate the conformational dynamics of $A_{2A}AR$. To preserve the native conformational

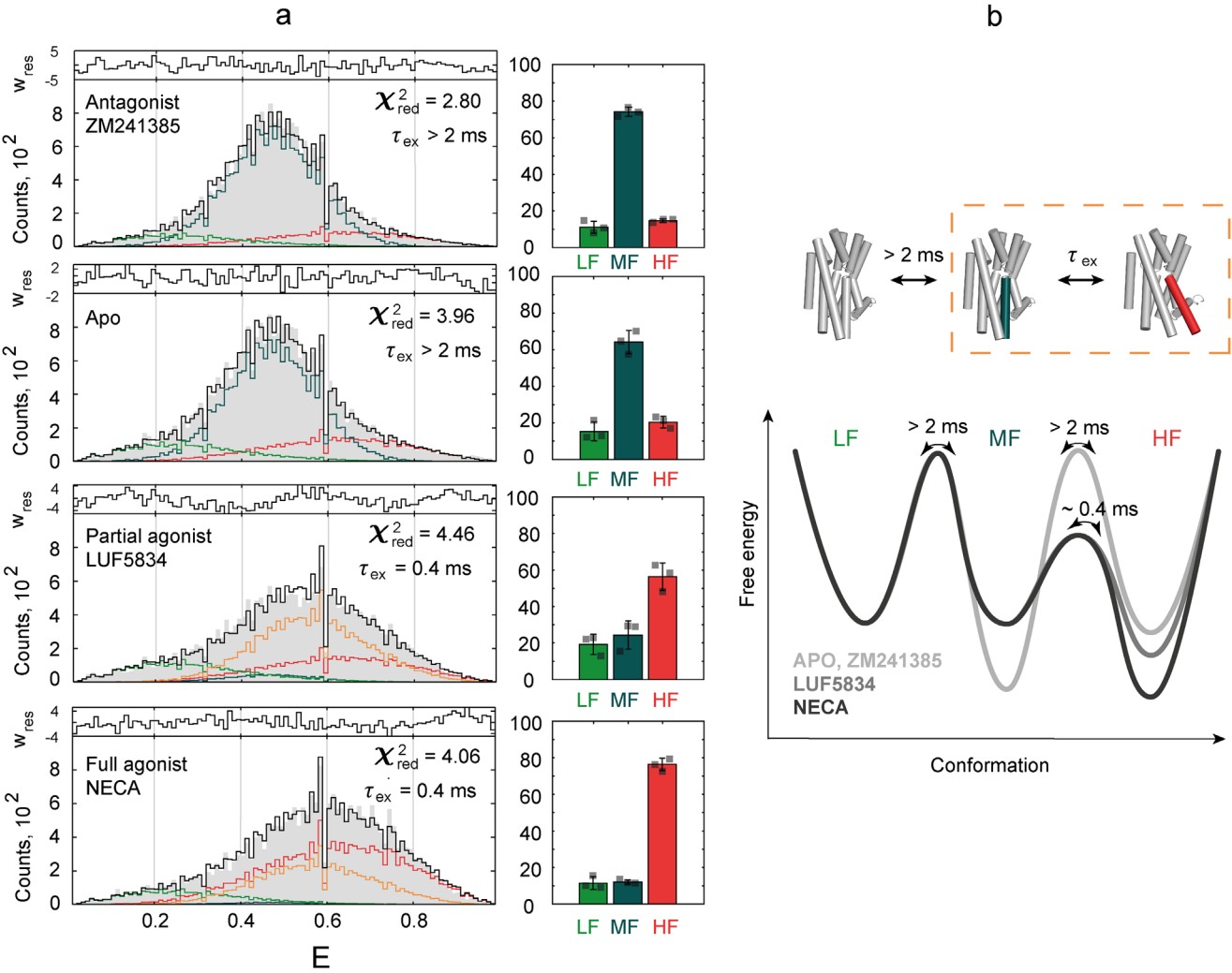

**Fig. 3 PDA quantifies parameters of the $A_{2A}AR$ three-state action model by fitting FRET efficiency distributions. a** Experimental distributions of 1-ms-long time bins derived from fluorescence bursts of double-labeled $A_{2A}AR$ (gray area) were fitted with a three-state model. The resulting fit (black line) is a sum of distributions simulated for molecules that stay in the LF (light green line), MF (dark cyan line), or HF (red line) state during the entire simulated time bin, and the distribution for molecules that sample both MF and HF states within the time bin (orange line). The fitting residuals are shown on the top of each panel. The bar charts on the right show relative populations of the three states, with error bars representing SD of $n = 3$ technical replicas with different protein aliquots. Individual data points are shown as gray squares, source data are available online as Supplementary Data 2. **b** The three-state action model of $A_{2A}AR$ and corresponding energy landscapes for the apo and agonist-bound receptor demonstrate relative populations of the states and inter-state exchange times. $\tau_{ex} = (k_{12} + k_{21})^{-1}$ is the relaxation time of the exchange between the MF and HF states (highlighted with dashed orange rectangle). TM6 is colored on the schematic (cylinder) representation of active (PDB: 5G53[70], red) and inactive (PDB: 3EML[69], dark cyan) structures of $A_{2A}AR$. The landscapes of relative energy are drawn with low FRET as a reference state.

dynamics of the receptor and minimize measurement-related artifacts we used four strategies. (1) As previous studies showed dramatic effects of commonly used detergents on GPCR conformational dynamics[64,65], we reconstituted the receptor in nanodiscs that provide a more relevant lipid bilayer environment. (2) To minimize the effect of fluorophores on the receptor's dynamics we used small organic dyes attached to strategically engineered cysteines. (3) We avoided the need to remove native cysteines and associated to that potential structural perturbations by using the previously developed in-membrane labeling procedure[52]. We showed that the mutant form of the receptor retains functional activity using the thermal shift assay and cAMP signaling assay in live HEK293T cells. (4) Finally, we studied receptors freely diffusing in solution and therefore excluded any artifacts related to their immobilization.

MFD-PIE fluorescence microscopy allowed us to measure and analyze FRET dynamics with a sub-millisecond temporal

resolution in the double-labeled receptor subpopulations while excluding unlabeled and single-dye labeled receptors from the analysis. Using various single-burst descriptors and time-resolved analysis methods for quantifying FRET dynamics, we revealed sub-millisecond conformational dynamics in $A_{2A}AR$. Slight deviations of bursts from the 'static FRET line' on the FRET efficiency versus donor fluorescence lifetime plot hinted at nanosecond-millisecond dynamics for the apo-$A_{2A}AR$ and $A_{2A}AR$ with each of the used ligands (Fig. 2b), although can equally be attributed to undefined systematic errors. FRET-2CDE analysis suggested more pronounced conformational dynamics in the agonist-bound $A_{2A}AR$ than in the apo or antagonist-bound $A_{2A}AR$ (Fig. 2d). BVA confirmed that the variations of FRET efficiency among ~100 μs time bins exceed the level expected from shot noise (Fig. 2f). Finally, fFCS clearly confirmed the dynamics nature of the data and demonstrated two components in $A_{2A}AR$ dynamics: fast microsecond-time (3–20 μs) dynamics

present in all samples and assigned mostly to dyes photophysics and slower (390 ± 80 µs) dynamics evoked by agonists (Fig. 2h). It is fFCS that puts all our findings in a single self-consistent picture: both fast and slow dynamics contribute to the deviation of bursts from the 'static FRET line', however the fast dynamics make limited contributions to the FRET-2CDE scores and to the BVA distribution deviations because of their 10-fold faster timescale compared to the temporal resolution of these techniques. Meanwhile, the slower dynamics evoked with the agonists explains the increased dynamics scores in FRET-2CDE and BVA for the agonist-bound $A_{2A}AR$. Finally, dynamic PDA led us to a three-state model of the $A_{2A}AR$ conformational dynamics that could fit the measured FRET efficiency histograms consistently with the fFCS findings (Fig. 3).

While our manuscript was under review, another publication appeared, showing active and inactive states in detergent-solubilized $A_{2A}AR$ via smFRET[66]. However, no agonist-induced increase in sub-millisecond dynamics were detected, which can be related to adverse effects of detergent environment on protein dynamics.

Our final three-state model of the $A_{2A}AR$ conformational space illustrates two effects of agonists on the $A_{2A}AR$: (i) the balance between MF and HF states is shifted away from the inactive-like MF towards the active-like HF, and (ii) the energy barrier between MF and HF states is lowered, as evidenced by more frequent transitions between the states (Fig. 3b). The MF and HF states are not interchangeable in the sub-millisecond time domain in the apo and antagonist-bound $A_{2A}AR$, but can interconvert on a 300–500 µs timescale in the agonist-bound $A_{2A}AR$. The least populated LF state, presumably, corresponds to receptors locked in a long-lived non-functional state or improperly folded receptors. In a good agreement with our findings, sub-millisecond agonist-induced conformational dynamics of the intracellular part of TM7[17] (Y290W[7.55]) and the N-terminal end of H8[20] (I292M[8.47]) have been shown for $A_{2A}AR$ by NMR. Although sub-millisecond dynamics between inactive-like conformations of TM6 (V229C[6.31]) in the apo-$A_{2A}AR$ have been reported[19], our data suggest only limited dynamics in the ligand-free and antagonist-bound $A_{2A}AR$.

The increased sub-millisecond conformational dynamics in the agonist-bound state may have a direct relevance to protein function. One of the most prominent structural changes associated with the agonist binding to $A_{2A}AR$ is the movement of the intracellular part of TM6 accompanied by a rotameric switch of Tyr197[5.58] side chain (PDB ID 3QAK[67] and PDB ID 2YDV[68]). In the antagonist-bound structure (PDB ID 3EML[69]), Tyr197[5.58] is placed between TM3 and TM6 forming a hydrogen bond network that tightens TM6 to the protein core. In the agonist-bound structure this residue moves outward, and the hydrogen bond network is lost. As a result, TM6 gains higher structural flexibility, an effect that we readily detected as increased sub-millisecond conformational dynamics in our experiments. In the further activation events upon G-protein binding to $A_{2A}AR$ (PDB ID 5G53[70]), the intracellular part of TM6 shifts further away from the receptor core and G-protein helix α5 protrudes into the created cleft. Therefore, the increased conformational dynamics may be important for the $A_{2A}AR$ to accommodate G protein and other signaling partners.

In our experiments, we could not measure slow conformational dynamics (>2 ms), because of the short residence time of individual molecules in the microscope focal spot. Our data do not indicate long-lived states in the agonist-bound $A_{2A}AR$ (besides the ligand-insensitive LF state), but the FRET-2CDE analysis shows only moderate dynamics scores, and the observed deviations from the 'static FRET line' can be explained by a microsecond plasticity within a long-lived conformation. Consequently,

we cannot exclude that long-lived conformations can coexist with conformations that show sub-millisecond dynamics. Keeping this in mind, we nevertheless did not introduce any additional long-lived states into our final model of the $A_{2A}AR$ conformational space to avoid overfitting. Previous studies based on NMR[19–21] and single-molecule fluorescence microscopy[32] provide complementary insights into the dynamics of long-lived (>2 ms) $A_{2A}AR$ conformations. Similarly to our study, both NMR[20] and single-molecule fluorescence microscopy[32] detect agonist-induced increase of the receptor conformational dynamics.

The observed agonist-induced increase in FRET efficiency is unexpected based on the distances between the labeled residues (L225 and Q310) in the available crystal structures of $A_{2A}AR$, which suggest a decrease in the FRET efficiency, because the distance between the Cα-atoms increases from ~40 Å in the antagonist-bound structure (PDB: 3EML[69]) to ~47 Å in the fully active structure (PDB: 5G53[70]). However, since the change of the distance between labeled residues (~7 Å) is smaller than the length of the flexible linkers attached to the dyes (~15 Å), we hypothesize that the inverse direction of the FRET change is due to the dyes not being randomly oriented (Fig. 1b), but rather occupy preferred locations within their respective accessible volume. In line with this, constrained dynamics of the dyes were indeed observed via fluorescence depolarization measurements: within nanoseconds after fluorescence excitation, anisotropy reached stable values ($r_p$) of 0.12–0.16 for the donor and 0.22–0.23 for the acceptor (Supplementary Fig. 13, Supplementary Table 7, "Fluorescence depolarization measurements" in "Methods"). These stable values correspond to wobbling within a cone with semi-angle of 43–49° or 34–35°, respectively, and are almost unaffected by ligands. To get further insights into the preferred locations of the dyes we performed 1-µs-long molecular dynamics (MD) simulations, which revealed that the dye attached to TM6 might indeed preferentially locate between the intracellular tips of TM3 and TM5 in the inactive conformation and enter the G-protein-binding cavity of the receptor in the fully active conformation (Supplementary Figs. 14–16, "Molecular dynamics simulations" in "Methods"). These preferred conformations of the dye would result in a significant decrease of the mean inter-dye distance upon receptor activation (from 5 nm to 3 nm, Supplementary Fig. 15), which would in turn lead to an increase in the mean FRET efficiency. Thus, our MD simulations provide a plausible explanation for the observed increase in the FRET efficiency upon $A_{2A}AR$ activation and show that the observed FRET changes agree with the available crystal structures of $A_{2A}AR$.

The PDA analysis of our data suggests that the partial agonist LUF5834 and the full agonist NECA stabilize 56 ± 8 % and 76 ± 3 % of $A_{2A}ARs$, respectively, in the same active-like HF conformation (mean ± SD, three technical replicas with different protein aliquots). A similar mechanism for partial agonism in $A_{2A}AR$ has recently been demonstrated by NMR with isotope-labeled methionine residues located in different structural domains (I106M[3.54], M140[4.61], M211[5.72], and I292M[8.47]) of the receptor[20]. On the other hand, other NMR-based studies have suggested that LUF5834 either stabilizes a distinct, not a fully active conformation[19,24], or has no effect on the $A_{2A}AR$ conformation[21]. Our data do not support the existence of a separate partially active conformation of $A_{2A}AR$ stabilized with LUF5834 that would be distinct from the fully active conformation stabilized with NECA. On the other hand, we cannot exclude that such partially active state could not be resolved in our data because small differences in FRET efficiency, the high photon shot noise in single-molecule experiments, or the broadening of the FRET-distribution due to variations of photophysical parameters of the dyes.

In addition, we observed that 20 ± 3% of apo $A_{2A}ARs$ exhibit an active-like HF state (Fig. 3a). It may be a reason for a moderate basal activity of the receptor, however the role the intracellular signaling partners cannot be excluded. Two previous NMR-based studies have addressed the molecular mechanisms of $A_{2A}AR$ basal activity. One study reported a 70% population of pre-active and fully active states in the apo-ensemble[19]. Another study has reported negligible basal activity and showed that in the agonist-bound $A_{2A}AR$ unique previously unpopulated conformations emerge[21]. A recent review suggests that discrepancies between these two works could arise from differences in used constructs, $^{19}F$ reporters, their attachment sites, or in selected membrane-mimicking systems (MNG/CHS versus DDM/CHS micelles)[71]. A recent NMR-based study with $A_{2A}ARs$ in nanodiscs reported a 50% population of active states in the the apo-ensemble[23]. The contradictory estimations of the basal activity of $A_{2A}AR$ should be put in the context of a similar heterogeneity of results provided by cell-based signaling assays. In different experiments, the basal activity of $A_{2A}AR$ was reported to reach from 0 to 20%[69,72–74], to 20–40%[46,75,76], or even 40–70%[77–80]. Cell assays are affected by different $A_{2A}AR$ expression levels and cell lines used[73,79]. It has been shown that the C-terminal truncation of $A_{2A}AR$ impairs its basal activity—this can play an important role for our study as well as previous NMR-based works[75].

Finally, our measurements show that ZM241385 does not change the distribution of FRET efficiency compared to apo conditions and therefore we do not observe inverse agonism of ZM241385. Because many studies reported negligible basal activity of $A_{2A}AR$, ZM241385 is widely referred to as $A_{2A}AR$ antagonist[11,69,81]. The recent $^{19}F$-NMR study, where no basal activity was detected for $A_{2A}AR$, correspondingly did not register any conformational changes induced with ZM241385[21]. On the other hand, those works that identified significant basal activity of $A_{2A}AR$ frequently reported inverse agonism of ZM241385[46,73,74,76,79,82]. In line with these findings, the $^{19}F$-NMR study that has reported 70% basal activity also showed inverse agonism of ZM241385[19]. Notably, it was previously shown that ZM241385 can loose inverse agonist activity if tested not in cells, but in isolated membranes[75]. This latter result suggests that intracellular interaction partners can play an important role in both basal activity and inverse agonism, explaining both heterogeneity in published functional data and our results.

The multi-state conformational behavior of GPCRs delineates their complex pharmacology and, therefore, challenges modern drug design. We believe that new methods showing how GPCR activity is modulated on a molecular level will facilitate the design and discovery of drugs with novel beneficial properties. Here we demonstrated a strategy to observe conformational dynamics of a GPCR in solution, yet in a close-to-physiological environment of lipid nanodiscs using intramolecular smFRET measured via the MFD-PIE approach. Our measurements combined fluorescence intensity, lifetime, and anisotropy information to characterize the sub-millisecond conformational dynamics of TM6 and H8 in $A_{2A}AR$ and shed light on molecular mechanisms of basal activity and partial agonism in the receptor. The general strategy developed in our work can be extended to study the effects of various modulators (ligands, ions, lipids, etc.), membrane-mimicking systems (micelles, lipid nanodiscs, liposomes, etc.) and genetic modifications on the activity of $A_{2A}AR$ and, in perspective, other GPCRs.

**Limitations of the study**. In this study, we used smFRET to investigate the dynamics of the $A_{2A}AR$. The intrinsic limitation of FRET as a label-based method is that the dynamics of dyes and protein cannot be completely separated based on fluorescence

data. Our nanosecond-time fluorescence depolarization measurements (Supplementary Fig. 13 and Supplementary Table 7) and microsecond-long MD simulations (Supplementary Figs. 14 and 15) indicate that the reorientation of the dyes attached to $A_{2A}AR$ upon conformational change of the protein strongly affects the measured FRET efficiency. This means that changes in FRET efficiency should not be interpreted exclusively as distance changes, and, particularly, apparent distances measured in PDA should only be considered as parameters of the fit, not as physical distances between the dyes. In addition, we cannot completely exclude that the dynamics of the dyes contribute to the observed 390-μs dynamics. However, fluorescence depolarization measurements suggest that the orientational freedom of the dyes is almost ligand independent (Supplementary Fig. 13 and Supplementary Table 7) and burst-wise anisotropy measurements do not indicate multiple long-lived states of the dyes on the millisecond timescale (Supplementary Fig. 17, "Burst-wise steady-state fluorescence anisotropies" in "Methods"). Therefore, we assign the agonist-induced dynamics observed in our data to the dynamics of the receptor. This interpretation is supported by previous NMR-based studies that also observed agonist-induced dynamics in $A_{2A}AR$ on a sub-millisecond timescale[17,20].

## Methods

**Protein expression, purification and labeling**. The gene encoding the human $A_{2A}AR$ (UniProt C9JQD8) C-terminally truncated after residue Ala 316 (Supplementary Fig. 1a) was synthesized de novo (Eurofins). The nucleotide sequence was optimized for *Leishmania tarentolae* expression with the GeneOptimizer software (ThermoFisher Scientific). KpnI restriction site was introduced at the C-terminus and used for polyhistidine tag (H9) fusion. The final construct was cloned into the integrative inducible expression vector pLEXSY_I-blecherry3 (Jena Bioscience, Germany) via the BglII and NotI restriction sites. L225C[6.27] and Q310C[8.65] mutations were introduced by PCR.

*Leishmania tarentolae* cells of the strain LEXSY host T7-TR (Jena Bioscience) were transformed with the $A_{2A}AR$ expression plasmids linearized by the SmiI restriction enzyme. After clonal selection, the transformed cells were grown at 26 °C in the dark in shaking baffled flasks in Brain-Heart-Infusion Broth (Carl Roth, Germany) supplemented with 5 μg/mL Hemin (AppliChem), 50 U/mL penicillin and 50 μg/mL streptomycin (both antibiotics from AppliChem). When $OD_{600} = 1$ was reached, 10 μg/mL tetracycline was added, and incubation continued for an additional 24 h.

The harvested cells were disrupted in an M-110P Lab Homogenizer (Microfluidics) at 10,000 psi in a buffer containing 50 mM $NaH_2PO_4/Na_2HPO_4$, pH 7.6, 0.2 M NaCl, 20 mM KCl, 10 mM $MgCl_2$, 10% glycerol (w/v), 1 mM EDTA, 2 mM 6-aminohexanoic acid (AppliChem), 50 mg/L DNase I (Sigma-Aldrich) and cOmplete protease inhibitor cocktail (Roche). The membrane fraction of the cell lysate was isolated by ultracentrifugation at $120,000 \times g$ for 1 h at 4 °C. The pellet was resuspended in the same buffer but without DNase I and stirred for 1 h at 4 °C. The ultracentrifugation step was repeated again.

Finally, the membranes were resuspended in the labeling buffer containing 50 mM HEPES, pH 7.0 10 mM $MgCl_2$, 20 mM KCl, 2 mM 6-aminohexanoic acid, and cOmplete and mixed with Atto643-maleimide (ATTO-TEC) and Alexa488 maleimide (Invitrogen), dissolved in dimethyl sulfoxide (0.5 mg of each fluorescent label per 10 g of cells). Labeling reactions were carried out overnight in the dark at 4 °C on a roller mixer.

The next day, membrane fractions were pelleted by ultracentrifugation at $120,000 \times g$ for 1 h at 4 °C and washed twice with the labeling buffer for removal of unbound fluorescent labels. For solubilization, membranes were resuspended in a buffer containing 20 mM HEPES, pH 8.0, 800 mM NaCl, 5 mM $MgCl_2$, 10 mM KCl, 2 mM 6-aminohexanoic acid, cOmplete with 4 mM theophylline (Sigma-Aldrich) and 1% n-Dodecyl β-maltoside (DDM) (Glycon Biochemicals)/0.2% cholesteryl hemisuccinate (CHS) (Merck) (w/v) and left on the stirrer for 2 h at 4 °C in the dark. The insoluble fractions were removed by ultracentrifugation at $120,000 \times g$ for 1 h at 4 °C. The supernatants were loaded on an Ni-NTA resin (Cube Biotech) and incubated in the batch mode overnight in the dark at 4 °C.

The next morning, proteins bound to Ni-NTA resin were washed with 10-column volumes of the first washing buffer: 50 mM HEPES, pH 7.5, 800 mM NaCl, 25 mM imidazole, 10 mM $MgCl_2$, 8 mM ATP (Sigma-Aldrich), 2 mM 6-aminohexanoic acid, 0.1 mM phenylmethylsulfonyl fluoride, 4 mM theophylline, cOmplete, 0.1 % DDM / 0.02% CHS. Then, columns were washed with 10-column volumes of the second washing buffer: 50 mM HEPES, pH 7.5, 800 mM NaCl, 50 mM imidazole, 2 mM 6-aminohexanoic acid, 0.1 mM phenylmethylsulfonyl fluoride, 4 mM theophylline, cOmplete, 0.1% DDM/0.02% CHS (w/v). Finally, proteins were eluted with 5-column volumes of the elution buffer: 25 mM HEPES, pH 7.5, 800 mM NaCl, 220 mM imidazole, 2 mM 6-aminohexanoic acid, 0.1 mM

phenylmethylsulfonyl fluoride, cOmplete, 0.1% DDM/0.02 % CHS (w/v). The eluates were subjected to size-exclusion chromatography on a Superdex 200 Increase 10/300 GL column (GE Healthcare Life Sciences) in a buffer containing 20 mM HEPES, pH 7.5, 800 mM NaCl, 1 mM EDTA, 2 mM 6-aminohexanoic acid, cOmplete, 0.05% DDM/0.01% CHS (w/v). Fractions, corresponding to $A_{2A}AR$ monomers, were pulled and subjected to nanodisc reconstitution.

**Nanodisc reconstitution**. Membrane Scaffold Protein 1D1 (MSP1D1) was expressed in *E. coli* using gene with an N-terminal 6XHis-tag and upstream TEV-protease site cloned into pET28a(+) (Addgene plasmid #20061[53]). MSP1D1 was purified using IMAC[83] with further cleavage of 6xHis-tag by TEV protease (Sigma-Aldrich). The lipid mixture of 1-palmitoyl-2-oleoyl-sn-glycero-3-phosphocholine (POPC): 1-palmitoyl-2-oleoyl-sn-glycero-3-phospho-(1'-rac-glycerol) (POPG) (Avanti Polar Lipids) in chloroform was prepared at a molar ratio 7:3. The lipid film was dried under a gentle nitrogen stream, followed by removal of the solvent traces under vacuum, and then solubilized in 200 mM sodium cholate. The purified $A_{2A}AR$ in DDM/CHS micelles was mixed with MSP1D1 and the POPC:POPG lipids at a molar ratio $A_{2A}AR$:MSP1D1:lipids = 0.2:1:60. The final sodium cholate concentration was adjusted to 20 mM, the typical receptor concentration was 0.1 mg/mL. After 1 h of incubation at 4 °C, the mixture was incubated with wet Bio-Beads SM-2 (Bio-Rad, 0.4 g of beads for 1 mL reaction, beads were washed in methanol and equilibrated with 20 mM HEPES, pH 7.5, 800 mM NaCl, 1 mM EDTA) overnight at 4 °C in the dark. The next morning, the beads were discarded and the supernatant was supplemented with a fresh portion of Bio-Beads for an additional 4 h incubation. Finally, $A_{2A}AR$ reconstituted into nanodiscs was sub-jected to size-exclusion chromatography on a Superdex 200 Increase 10/300 GL column (GE Healthcare) in a buffer containing 20 mM HEPES, pH 7.5, 150 mM NaCl, 1 mM EDTA, 2 mM 6-aminohexanoic acid, cOmplete. Labeling efficiencies of 26% (Alexa488) and 8% (Atto643) were obtained for the mutant $A_{2A}AR$ (Supplementary Table 1). The low labeling efficiency is, probably, a consequence of labeling the receptors directly in isolated membrane pellets, a strategy we adopt to avoid mutating out native cysteines protected by the native lipid bilayer from labeling. Fractions containing labeled receptors were combined together and used for further experiments.

**Fluorescence spectra characterization**. For fluorescence spectra characterization, diluted (<5 μM) apo-$A_{2A}AR$ samples were placed in a quartz cuvette (10 mm path length). Excitation and emission spectra were recorded using an Edinburgh Instruments FLS980 spectrometer corrected for the wavelength-dependent throughput and sensitivity of the detector. Fluorescence in the acceptor's emission spectral range after irradiation in the donor's excitation spectral range indicated FRET in the double-labeled receptor samples (Supplementary Fig. 1c).

**Ensemble-based fluorescence lifetime measurements**. The time-resolved detection of the fluorescence decay of Apo-$A_{2A}AR$ labeled with Alexa488 and Atto643 was performed with a Fluotime100 fluorescence spectrophotometer (Picoquant, Berlin, Germany) based on a picoHarp300 unit and using a pulsed diode laser (LDH-440; center wavelength 440 nm; pulse width 54 ps; repetition frequency 10 MHz) as an excitation source. Fluorescence decay curves were measured at 665 nm under magic angle conditions by time-correlated single-photon counting (TCSPC) allowing to determine fluorescence lifetimes down to 100 ps[84]. Decay curves were analyzed by iterative reconvolution of the instrument response function, IRF(t), with an exponential model function, M(t), using the FluoFit software (version 4.4; Picoquant).

Fitting the measured TCSPC-delay signal with a monoexponential decay (Supplementary Fig. 1d) did not allow a satisfactory description of the acceptor fluorescence intensity time trace, while a biexponential fit was sufficient and yielded two components: one with a positive amplitude (normal fluorescence decay) and one with a negative amplitude (rise term). The rising term is expected for FRET and cannot originate exclusively from the direct excitation of Atto643 with a 440-nm laser. Therefore, fluorescence lifetime measurements of labeled mutant protein in bulk solution also confirmed that there is a fraction of double-labeled receptors that exhibit FRET in the sample.

**Thermal shift assay**. To show that the $A_{2A}AR$ mutant (L225C[6.27]/Q310C[8.65]) retains ligand-binding activity in lipid nanodiscs, we used the fluorescent thermal stability assay[54]. The studies were carried out on a Rotor-Gene Q 6 plex (QIAGEN) instrument at a heating rate of 2 °C/min and a temperature range of 25–90 °C. The excitation wavelength was set at 387 nm and the emission wavelength was 463 nm. The $A_{2A}AR$ concentration was about 2 μM. Buffer conditions: 20 mM HEPES, 150 mM NaCl, 1 mM EDTA, 2 mM 6-aminohexanoic acid, pH 7.5. To obtain a good fluorescence intensity we used a 2.5-fold molar excess of CPM dye (7-Diethylamino-3-(4'-Mal-eimidylphenyl)-4-Methylcoumarin, Invitrogen) to protein. To prepare protein for the ligand-binding measurements, we added 200 μM of ZM241385 or NECA and incu-bated for 1 h in the dark at 4 C. The thermal denaturation assay was performed in a total volume of 50 μL (Supplementary Fig. 1e).

**Measurement of $A_{2A}AR$ surface expression and Gs-signaling**. For $A_{2A}AR$ functional assays, the $A_{2A}AR$ (WT or L225C[6.27]/Q310C[8.65] mutant, both

C-terminally truncated after residue Ala 316) gene (GenScript) was optimized for eukaryotic expression with an N-terminal hemagglutinin signal sequence (MKTIIALSYIFCLVFA) followed by the FLAG tag epitope (DYKDDDDK) and C-terminal 10×His tag were cloned into pcDNA3.1(-) at BamHI(5′) and HindIII(3′). The surface expression of $A_{2A}AR$ was determined by the whole-cell ELISA assay[85]. Briefly, HEK293FT cells were seeded in a 100 mm cell culture plate and transfected separately with 10 μg of each expression plasmid DNA (pcDNA3.1(-)_$A_{2A}AR$(WT), pcDNA3.1(-)_$A_{2A}AR$(L225C[6.27]/Q310C[8.65]) or pcDNA3.1(-) as a negative control) using a common Lipofectamine 3000 protocol. The plates were incubated for an additional 12–18 h at 37 °C, 5% CO₂. The HRP-conjugated anti-FLAG M2 antibody (A8592, Sigma) at a dilution of 1:2000 in TBS with 1% protease-free BSA (A3059, Sigma) and TMB ready-to-use substrate (T0565, Sigma) were used for the ELISA procedure. For normalization on cells quantity Janus Green B (Sigma) staining was used, and the absorbance ratio $A_{450}/A_{595}$ was calculated. Measurements were per-formed in triplicate for WT and mutant $A_{2A}AR$ as well as for empty-vector-transfected cells. Measured values of $A_{450}/A_{595}$ were normalized so that the mean expression level of WT $A_{2A}AR$ was 100% ($F_{WT} = 100 ± 6\%$, SDs for $n = 3$ mea-surements are given). The double-mutant form of the receptor showed only slightly lower expression level than WT: $F_{L225C/Q310C} = 73 ± 7\%$. Empty-vector-transfected cells showed only marginal anti-FLAG antibody binding: $F_{EV} = 1 ± 1\%$.

For evaluation of the $A_{2A}AR$ signaling activity, we checked the effect of the agonist NECA on cAMP responses in transfected cells. For cAMP determination, we used the Bioluminescence Resonance Energy Transfer (BRET) approach with the EPAC biosensor[86]. The cAMP BRET biosensor was kindly provided by professor Raul Gainetdinov[87]. Transfections were carried out with Lipofectamine 3000 (Thermo) using HEK293T cells seeded in a 100 mM cell culture plate, receptor cDNA vectors pcDNA3.1(-)_$A_{2A}AR$(WT, residues 1–316), pcDNA3.1(-) _$A_{2A}AR$(Q310C[8.65]/L225C[6.27], residues 1–316) or empty pcDNA3.1(-) vector (10 μg each) and the EPAC biosensor cDNA vector (1 μg) needed for evaluation of the cAMP production. Transfected cells were split into 96-well plates at $10^5$ cells per well. On the following day, 70 μL of PBS were added to each well followed by the addition of 10 μL of a 50 μM coelenterazine-h solution (Promega). After 10-min incubation, either 10 μL of buffer or 10 μL of NECA at different concentrations in PBS were added, and the plate was then placed into a CLARIOstar reader (BMG LABTECH, Germany) with a special BRET filter pair (475 ± 30 nm—coelenterazine-h and 530 ± 30 nm—YFP). The BRET signal was calculated as the ratio of the light emitted at 530 nm to the light emitted at 480 nm. Three independent experiments with three technical replicas in each were conducted. For pEC50 evaluation, dose-response curves from three technical replicas were averaged and analyzed. Mean and SD of pEC50 among three biological samples were calculated (Supplementary Fig. 1f). A lack of agonist-induced BRET changes in cells transfected with an empty vector confirmed that signaling from the endogenous $A_{2A}AR$ in HEK293T cells is negligible.

**Confocal MFD-PIE setup**. For single-molecule experiments, a home-built multi-parameter fluorescence detection microscope with pulsed-interleaved excitation (MFD-PIE)[39] was used (see scheme of the setup in Supplementary Fig. 2). Two lasers were used: a pulsed 483-nm laser diode (LDH-P-C-470, Picoquant) and a pulsed 635-nm laser diode (LDH-P-C-635B, Picoquant), with alternating at 26.67 MHz pulses, delayed by 18 ns with respect to each other. Sample emission was transmitted through a pinhole and spectrally split. Both, the blue range and red range were split by polarization into two detection channels. Photons were detected by four avalanche photodiodes (PerkinElmer or EG&G SPCM-AQR12/14, or Laser Components COUNT BLUE): $B_{||}$ (blue-parallel), $B_{\perp}$ (blue-perpendicular), $R_{||}$ (red-parallel) and $R_{\perp}$ (red-perpendicular) (Supplementary Fig. 2), which were connected to a TCSPC device (SPC-630, Becker & Hickl GmbH). Microscope alignment (excitation light guiding, objective lens correction collar, pinhole, detectors) was done using real-time fluorescence correlation spectroscopy (FCS) on freely dif-fusing Atto488-COOH and Atto655-COOH in water. For more details about the used equipment the reader is referred to ref. [88].

**smFRET data recording**. Samples of double-labeled $A_{2A}AR$ in nanodiscs were diluted in a buffer, containing 20 mM HEPES, pH 7.5, 150 mM NaCl, 1 mM EDTA, 2 mM 6-aminohexanoic acid to a protein concentration of 0.5-2 nM. To measure the effects of ligand binding, samples were supplemented with either 10 μM ZM241385, 10 μM LUF5834 or 10 μM NECA and incubated for 30 min at 4 °C. After the incubation, the samples were transferred to a Nunc Lab-Tek Chambered coverglass (Thermo). smFRET experiments were performed at 100 μW of 483 nm and 50 μW of 635 nm excitation. Measurements were recorded at room temperature (22 °C), samples were replenished every 30 min. With all filters applied (see *Selection of double-labeled, donor-only and acceptor-only subpopulations*), 9000–12,000 bursts corresponding to double-labeled molecules were collected for each sample: 11,961 for apo, 10,167 burst for ZM241385, 9557 for LUF5834, and 11,007 for NECA. Background scattering information was obtained via a buffer measurement under identical conditions.

**Burst identification**. For single-molecule data, we employed a two-color MFD all-photon burst search algorithm[63] using a 500-μs sliding time window (min. 50

photons per burst, min. five photons per time window). A 0–20-ms burst duration cutoff was applied to remove sparse (<1%) slow-moving aggregates.

**Selection of double-labeled, donor-only, and acceptor-only subpopulations.** To select single-labeled or double-labeled subpopulations of molecules, we used specific restrictions for the stoichiometry $S$, FRET efficiency $E$, fluorescence lifetime, anisotropy, and kernel-density estimator ALEX-2CDE, as shown below.

Donor-only molecules: $ALEX\text{-}2CDE > 20$, $-0.1 < E < 0.1$, $0.9 < S < 1.1$, $0.1$ ns $< \tau_D < 6$ ns, $-0.2 < r_D < 0.6$.

Acceptor-only molecules: $ALEX\text{-}2CDE > 20$, $0.6 < E < 1.1$, $-0.1 < S < 0.2$, $0.1$ ns $< \tau_A < 8$ ns, $-0.2 < r_A < 0.6$.

Double-labeled molecules: $ALEX\text{-}2CDE < 15$, $0.1 < E < 1.0$, $0.2 < S < 0.8$, $0.1$ ns $< \tau_D < 4.5$ ns, $0.1$ ns $< \tau_A < 8$ ns, $-0.2 < r_D < 0.6$, $-0.2 < r_A < 0.6$.

**FRET efficiency and stoichiometry.** The absolute burst-averaged FRET efficiency E was calculated as:

$$E = \frac{F_{BR} - ct \cdot F_{BB} - de \cdot F_{RR}}{\gamma F_{BB} + F_{BR} - ct \cdot F_{BB} - de \cdot F_{RR}} \qquad (1)$$

where $F_{BR} = S_{BR} - B_{BR}$ is the background-corrected number of photons in the red detection channels independently of the polarization after blue excitation (with $S_{BR}$ and $B_{BR}$ being the summed intensity and background, respectively, in the time gates $BR_{\parallel}$ and $BR_{\perp}$); $F_{BB} = S_{BB} - B_{BB}$ is the background-corrected number of photons in the blue detection channels after blue excitation (with $S_{BB}$ and $B_{BB}$ being the summed intensity and background, respectively, in the time gates $BB_{\parallel}$ and $BB_{\perp}$), $F_{RR} = S_{RR} - B_{RR}$ is the background-corrected number of photons in the red detection channels after red excitation (with $S_{RR}$ and $B_{RR}$ being the summed intensity and background, respectively, in the time gates $RR_{\parallel}$ and $RR_{\perp}$), $de$—the correction factor for direct excitation of the acceptor with the 483 nm laser, $ct$—the correction factor for the emission crosstalk of the donor in the acceptor channel, and $\gamma$—the relative detection efficiency of the donor and acceptor[39].

The corrected stoichiometry ratio $S$ was calculated with:

$$S = \frac{\gamma F_{BB} + F_{BR} - ct \cdot F_{BB} - de \cdot F_{RR}}{\gamma F_{BB} + F_{BR} - ct \cdot F_{BB} - de \cdot F_{RR} + \beta F_{RR}}, \qquad (2)$$

where $\beta$-factor accounts for different detection efficiencies of the donor and acceptor.

**Correction factors.** For correction, first, the background was subtracted from the experimental signals. Then, the donor emission crosstalk ($ct = 0.0059$) and acceptor direct excitation ($de = 0.024$) factors were determined directly from the measurements and applied to correct the data[39]. For correction purposes, we preliminarily (see the final selection criteria for other analyses in the section "Selection of double-labeled, donor-only and acceptor-only subpopulations") selected double-labeled molecules using the kernel-density estimator (ALEX-2CDE < 15)[42], FRET efficiency ($0.1 < E < 1$), and stoichiometry ($0.2 < S < 0.6$), corrected for channel crosstalk ($ct$) and direct excitation ($de$). For these selected molecules, $E$ was plotted vs $1/S$, and a straight line was fitted to obtain the correction factors:

$$\gamma = \frac{\Omega - 1}{\Omega + \Sigma - 1} \qquad (3)$$

$$\beta = \Omega + \Sigma - 1, \qquad (4)$$

where $\Omega$ is the intercept and $\Sigma$ is the slope of the fit. Finally, $\gamma = 0.69$ and $\beta = 1.9$ were obtained.

**Burst-wise fluorescence lifetime.** To estimate the single-molecule burst-averaged fluorescence lifetimes of the donor ($\tau_D$) and acceptor ($\tau_A$), the maximum likelihood estimator approach was used[89].

To test whether the deviations of bursts from the static FRET line are statistically significant, experimental measurements were supplemented with simulations. Experimental bursts were grouped in $N = 50$ intervals equally spaced by the burst-wise FRET efficiency. Groups with less than 200 bursts were excluded from the analysis.

Then, for each burst a distance $R_i(E_i)$ was determined so that a simulated normal distance distribution with the linker-related standard deviation $\sigma = 6$ Å centered around $R_i$ provided a mean FRET efficiency $E_i$ given the Förster distance $R_0 = 49$ Å.

Within a burst, for each photon in the donor channel, one distance $R_{ij}$ was simulated from the normal distribution with a mean distance $R_i$ and a standard deviation $\sigma$. For each generated distance, corresponding theoretical FRET efficiency ($E_{ij}(R_{ij})$) and fluorescence lifetime ($\tau_{ij}(R_{ij})$) were calculated. Also, for each photon, a simulated photon arrival time ($t_{ij}$) was generated as a random variable with exponential distribution given the theoretical fluorescence lifetime ($\tau_{ij}(R_{ij})$).

The intensity-weighted fluorescence lifetime ($\tau_{INT}$) was calculated across all photons belonging to all bursts within a group with ($1-E_{ij}$) weights:

$$\tau_{INT} = \frac{\sum\limits_{i,j} t_{ij}(1 - E_{ij})}{\sum\limits_{i,j}(1 - E_{ij})} \qquad (5)$$

The simulation was repeated 100 times. The confidence interval for the intensity-weighted donor lifetime was calculated using the standard deviation across the simulated values, significance level $\alpha = 0.001$, and Bonferroni correction for multiple groups. Finally, we compared the experimental intensity-weighted donor lifetime $\tau_{INT}$ with the upper border of the confidence interval (Supplementary Fig. 4).

This approach was established in the PAM software and the code is available online[90].

**FRET-2CDE analysis.** The FRET-2CDE scores for individual bursts were calculated with the time-kernel 100 μs as described previously[42].

To test whether the deviations of bursts from the static FRET-2CDE line were statistically significant, experimental bursts were grouped in $N = 50$ intervals equally spaced in the burst-wise FRET efficiency. Groups with less than 200 bursts were excluded from the analysis.

Then we run a simulation, in which photons in each burst were randomly "recolored" with donor and acceptor channel probabilities corresponding to the burst-wise FRET efficiency. Weighted FRET-2CDE values were calculated for each group:

$$FRET - 2CDE = 110 - 100 \cdot \left( \frac{\sum N_D E_D}{\sum N_D} + \frac{\sum N_A (1 - E)_A}{\sum N_A} \right), \qquad (6)$$

where $N_D$ and $N_A$ are numbers of donor and acceptor photons per burst, and $E_D$ and $(1-E)_A$ were calculated for each burst as described in ref. [42]. The simulation was performed 1000 times. The mean weighted FRET-2CDE and the 99.9% confidence intervals (with Bonferroni correction) were calculated for each group. Weighted FRET-2CDE values observed in $A_{2A}AR$ data exceed the calculated confidence intervals (Supplementary Fig. 5).

**Burst variance analysis (BVA).** BVA was performed as described[43]. Each fluorescence burst was segmented into $M_i$ bins of $n = 5$ consecutive photons; the proximity ratio $\epsilon_{ij}$ was calculated for each bin by the ratio $N_a/n$, where $N_a$ is the number of acceptor photons within the bin. The burst-wise proximity ratio $PR_i$ was calculated for each burst by the ratio $N_a/N$, where $N$ is the total number of photons within the burst and $N_a$ is the number of acceptor photons within the burst. From the resulting set $\{\epsilon_{ij}\}$ and the burst-wise proximity ratio $PR_i$, the standard deviation is estimated as:

$$s_i = \sqrt{\frac{1}{M_i}\sum\limits_{j=1}^{j=M_i}\left(\epsilon_{ij} - PR_i\right)^2}. \qquad (7)$$

The burst-wise $s_i$ values were plotted against burst-wise FRET efficiency.

Bursts were grouped in $N = 20$ equally spaced intervals by the burst-wise proximity ratio $PR_i$; only groups with >100 bursts were analyzed. Within each group, the mean value of $\{\epsilon_{ij}\}$ was determined, and the corresponding FRET efficiency value was calculated using correction factors $ct$, $de$, $\gamma$, and $\beta$ (eq. 29 from ref. [91]):

$$E = \frac{1 - (1 + ct + \gamma\beta \cdot de)(1 - PR)}{1 - (1 + ct - \gamma)(1 - PR)} \qquad (8)$$

The standard deviation of $\{\epsilon_{ij}\}$ within each group was plotted against FRET efficiency.

For comparison, the theoretical 'static' standard deviation $s$ was determined:

$$s = \sqrt{\frac{PR(1 - PR)}{n}}. \qquad (9)$$

The 99.9% confidence interval for $s$ was determined from simulated "static" bursts, given the same number of bursts in each group. The theoretical "static" standard deviation and confidence intervals were plotted against corrected FRET efficiency (Fig. 2d).

**Filtered fluorescence correlation spectroscopy (fFCS).** The mathematical background of fFCS was described in detail[92]. We built two reference TCSPC patterns corresponding to the "low-FRET" pseudo-species ($p_j^{LF}$) and "high-FRET" pseudo-species ($p_j^{HF}$) (Supplementary Fig. 7). For this, we merged all four smFRET datasets with different ligand conditions; bursts corresponding to double-labeled receptors with $E < 0.3$ were used to build $p_j^{LF}$, bursts corresponding to double-labeled receptors with $E > 0.7$ were used to build $p_j^{HF}$. TCSPC channels for $BB_{\parallel}$, $BB_{\perp}$, $BR_{\parallel}$, and $BR_{\perp}$ excitation and emission channels were stacked into a single array and indexed with $j$ for global analysis. Using the reference TCSPC patterns, $p_j^{LF}$ and $p_j^{HF}$ filters $f_j^{LF}$ and $f_j^{HF}$ were calculated as described[92]. To reduce noise in fFCS filters, at this step TCSPC bin was increased to 100 μs.

Using the reference filters $f_j^{LF}$ and $f_j^{HF}$ and the fluorescence signal $S_j$, the correlation function $G(\tau)$ was calculated for each dataset:

$$G(\tau)^{(i,m)} = \frac{<(\sum_{j=1}^{C} f_j^{(i)} S_j(t)) \cdot (\sum_{j=1}^{C} f_j^{(m)} S_j(t))>}{<(\sum_{j=1}^{C} f_j^{(i)} S_j(t))> \cdot <(\sum_{j=1}^{C} f_j^{(m)} S_j(t))>} - 1. \quad (10)$$

Only bursts from double-labeled molecules were taken into account; a 10 ms time window was introduced to reduce artifacts related to the sub-ensemble FCS analysis.

The cross-correlation function $G^{LF,HF}$ was fit using equation:

$$G^{(LF,HF)}(\tau) = G_{diff}(\tau)\left(1 - A_1 e^{-\frac{\tau}{\tau_1}} - A_2 e^{-\frac{\tau}{\tau_2}}\right) + y_0, \quad (11)$$

where the diffusion-limited term is:

$$G_{diff}(\tau) = \frac{1}{\sqrt{8}N} \frac{1}{(1 + \tau/\tau_{diff})(1 + \tau/p^2\tau_{diff})^{1/2}}. \quad (12)$$

The resulting cross-correlation curves were normalized using $N$ and offset $y_0$ and plotted in Fig. 2h. The 95% confidence intervals for the fitting parameters were calculated using the numerical Jacobian matrix.

**Photon distribution analysis (PDA).** Dynamic PDA was carried out to quantify the populations of FRET states and account for the conformational dynamics revealed by other analysis approaches[62,93]. Practically, for each smFRET dataset, raw bursts were re-binned in different time bins (0.5, 1, and 2 ms), and three histograms were constructed and analyzed simultaneously. Only bins with at least 20 and maximally 300 photons (to reduce calculation time) were further analyzed using PDA. Bins with uncorrected stoichiometry $S_{PR}$ below 0.2 or above 0.6 were removed from the analysis, because of suspected complex acceptor photophysics or photobleaching. Correction parameters $\gamma = 0.69$, $ct = 0.0059$ and $de = 0.024$, as well as the average background count rates in the donor and the acceptor channels after donor excitation were used to calculate the corrected FRET efficiency for PDA. The mean and width of all Gaussian distributed sub-states were globally optimized over all (three ligand and apo) conditions. State areas $A_i$ for static states and interconversion rates constants $k_{12}$ and $k_{21}$ for dynamic states were optimized for each sample. An fFCS-constrained PDA was performed with a fixed exchange time $\tau_{ex} = (k_{12} + k_{21})^{-1}$ and $k_{12}/k_{21}$ ratio optimized globally for each sample. The exchange time was fixed to values determined from fFCS ($\tau_{ex} = \tau_2$) for agonists, and to virtual infinity (>100 ms) for the antagonist-bound or apo receptors (to account for reduced amplitude of dynamic term). Experimental corrected FRET efficiency histograms were fitted using a reduced $\chi^2$-guided simplex search algorithm. The resulting parameters are presented as the means ± SD of three technical replicas with different protein aliquots in Supplementary Fig. 12 and Supplementary Table 6. For the two-state PDA, the population of the static state $A_3$ was set to zero (Supplementary Fig. 10 and Supplementary Table 4). For the three-state static PDA, fit was performed with $\tau_{ex}$ set to virtual infinity (>100 ms) for apo and all ligand-bound conditions (Supplementary Fig. 11 and Supplementary Table 5). Criteria for a good fit were a low (<4) global Poissonian $\chi_{red}^2$ value.

In our data, FRET efficiency distributions for MF and HF states are wide and overlapping. Thus, a low-FRET contrast led to a low sensitivity of PDA for dynamics in the data. To test how well our final fFCS-constrained PDA model describes the experimental data compared to the three-state static model, we independently analyzed three datasets, each dataset was obtained from a separate protein aliquot and contained 2000-5000 bursts. We treated these three datasets independently and calculated the mean $\chi_{red}^2$ with SD for fully static and fFCS-constrained models. The static model fitted the reduced $\chi_{red}^2 = 1.7 \pm 0.1$, while the fFCS-constrained model resulted in $\chi_{red}^2 = 1.9 \pm 0.3$. Therefore, PDA $\chi_{red}^2$ cannot distinguish between the three-state static and fFCS-constrained models.

**Fluorescence depolarization measurements.** The setup described in "Methods" "Confocal MFD-PIE setup" was used for fluorescence depolarization measurements albeit after improving the temporal resolution of photon detection. Particularly, detectors in the donor channels were replaced with single-photon avalanche diodes (Picoquant MPD PDM-100-CTE, < 25 cps). To preserve the timing resolution, the NIM output of the donor detectors was used. NIM-to-TTL converters (NIM2TTL, Micro Photon Devices) were used to connect NIM outputs of the donor detectors to the photon router (Becker-Hickl HRT82). Measurements were done as described in "Methods" "smFRET data recording", updated correction factors were applied: $ct = 0.018$, $de = 0.024$, $\gamma = 0.97$, $\beta = 2.1$, $G_B = 0.95$, $G_R = 1.04$. Bursts corresponding to single-labeled molecules were selected. Time-resolved fluorescence anisotropy $r(t)$ was calculated as

$$r(t) = \frac{GF_{\parallel}(t) - F_{\perp}(t)}{GF_{\parallel}(t) + 2F_{\perp}(t)}, \quad (13)$$

where $F_{\parallel}(t)$ is the intensity in the time gate $BB_{\parallel}$ (donor) or $RR_{\parallel}$(acceptor), and $F_{\perp}(t)$ is the intensity in the time gate $BB_{\perp}$ (donor) or $RR_{\perp}$ (acceptor). Experimental anisotropy decays $r(t)$ (Supplementary Fig. 13) were fitted with a biexponential model:

$$r(t) = \left((r_0 - r_p)e^{-t/\rho_F} + r_0\right)e^{-t/\rho_P}. \quad (14)$$

In all cases $\rho_p$, which describes slow depolarization due to rotation of the protein as a whole, was >50 ns, and therefore affected the fitting process only slightly. Three technical replicas with different protein aliquots were performed, mean and SD values for fitting parameters $r_0$ (fundamental anisotropy), $r_p$ (residual anisotropy), $\rho_F$ (fluorophore relaxation time) are given in Supplementary Table 7.

Finally, we used a "wobbling-in-a-cone"[94] model to calculate the typical angular displacement of the fluorophores:

$$\frac{r_p}{r_0^*} = \left(\frac{\cos\theta(1 + \cos\theta)}{2}\right)^2, \quad (15)$$

where $r_0^* = 0.4$ is the fundamental fluorescence of the dyes. We attribute the difference between $r_0^*$ and the observed $r_0$ to fast rearrangement of the dyes that is not detectable due to the temporal resolution of the apparatus.

**Molecular dynamics simulations.** The initial model of the $A_{2A}AR$ in the inactive state (amino acids 3–316) embedded in the membrane was prepared using the CHARMM-GUI web-service[95] based on the structure of a thermostabilized $A_{2A}AR$ in complex with ZM241385 (PDB ID: 3PWH)[82]. The thermostabilized mutations were mutated back to native amino acids and the missing regions were added using MODELLER[96] with an exception of the loop 212–223, which was omitted to prevent possible interference with the fluorescent label at the position 225 and thus improve its sampling. The structure of $A_{2A}AR$ in complex with mini-Gs (PDB ID 5G53)[70] including residues 309–312 from the C-terminal linker was used as a template to model the wild-type variant C-terminal residues missing in 3PWH (residues 306–316) and 5G53 (residues 309–316) with residue 310 substituted with a cysteine. The Atto647N-maleimide and Alexa488-C5-maleimide fluorescent labels were attached at the positions 225 and 310 by aligning the backbone atoms of the modified cysteine residues with the bound fluorescent labels to the backbone atoms of corresponding residues of the protein. In total, six simulations were performed: three for each double-labeled variant of $A_{2A}AR$. In these simulations, we used the Atto647N-maleimide dye instead of its derivative Atto643-maleimide used in the experiment, because the structure of the latter was not published. The resulting solvated systems contained 76,765 atoms including 203 POPC lipids, 68 sodium, and 75 chloride ions (labeling variant 1 with Atto647N attached to $L225C^{6.27}$, and Alexa488 attached to $Q310C^{8.65}$) and 76,810 atoms including 203 POPC lipids, 68 sodium, and 75 chloride ions (label variant 2 with Alexa488 attached to $L225C^{6.27}$, and Atto647N attached to $Q310C^{8.65}$). The simulation boxes had total dimensions of $9.02 \times 9.02 \times 9.20$ nm$^3$ and $9.01 \times 9.01 \times 9.19$ nm$^3$, respectively. All ionizable amino acids were modeled in their standard ionization state at pH 7. The CHARMM-GUI recommended protocol was applied for the initial energy minimization and equilibration of the system. During all of the equilibration steps, the force constants of the harmonic positional restraints on lipids were gradually reduced to zero while those on the protein Cα-atoms were left intact.

The equilibration simulations were followed by targeted MD simulations[97] in order to steer the systems to the fully active state while inducing minimal effects on the systems. The 5G53 structure was used as a target for targeted simulations to the fully active state. For the targeted MD simulations, the Nose–Hoover thermostat and the Parrinello–Rahman barostat were used. The temperature and pressure were set to 313.3 K and 1 bar with temperature and pressure coupling time constants of 1.0 ps$^{-1}$ and 0.5 ps$^{-1}$, respectively. Each targeted simulation was run for 100 ns with the force constant of 50,000 kJ/mol applied to the protein Cα-atoms only.

The triplicate production simulations for the inactive and fully active states were run for 1000 ns in NVT ensemble (maintained by the Nose–Hoover thermostat with $T_{ref} = 313.3$ K, temperature coupling time = 1.0 ps$^{-1}$) with the protein Cα-atoms constrained by harmonic potentials (1000 kJ/mol/nm$^2$). Each individual simulation was additionally prefaced by a short (10 ns, excluded from the further analysis) equilibration simulation with the random velocities drawn from Maxwell distribution to guarantee the independence of initial conformations. The fluorescent labels were coupled separately to a heat bath ($T_{ref} = 450$ K) to enhance conformational sampling[98].

All MD simulations were performed by GROMACS version 2020.2[99] with the PLUMED plugin[100] used for the targeted MD. A time step of 2 fs was used for equilibration simulations except for the early steps (where it was 1 fs), while targeted and production simulations were performed with a 4-fs time step allowed by repartitioning the mass of heavy atoms into the bonded hydrogen atoms[101] and the LINCS constraint algorithm[102]. The CHARMM36m force field was used for the protein, lipids, and ions[103]. The topologies for the fluorescent labels were obtained using the CGenFF web-service version 1.0.0 (force field version 3.0.1)[104]. They are available online as Supplementary Data 3 and Supplementary Data 4.

In order to estimate the convergence of label sampling, we calculated the volume available for each label as a function of simulation time (Supplementary Fig. 16). The estimation was done using the custom script available at https://github.com/porekhov/A2a_smFRET.

All trajectories of the production simulations are available at https://github.com/porekhov/A2a_smFRET.

**Burst-wise steady-state fluorescence anisotropies**. Burst-wise steady-state fluorescence anisotropies of the donor ($r_D$) and the acceptor ($r_A$) were calculated from the respective fluorescence intensities:

$$r = \frac{G F_{\parallel} - F_{\perp}}{G F_{\parallel} + 2 F_{\perp}} \qquad (16)$$

where $G$ is the correction factor for different detection efficiencies in the two polarization channels ($G_B = 0.99$, $G_R = 1.13$), $F_{\parallel}$ is the intensity in the time gate $BB_{\parallel}$ (donor) or $RR_{\parallel}$ (acceptor), and $F_{\perp}$ is the intensity in the time gate $BB_{\perp}$ (donor) or $RR_{\perp}$ (acceptor).

**Statistics and reproducibility**. smFRET data were collected for three different protein aliquots and all data derivatives are given as the mean ± SD. For the cross-correlations fFCS functions, error bars are calculated as SDs obtained after splitting the photon data into ten equally sized bines (Supplementary Fig. 9); the 95% confidence intervals for the fitting parameters were calculated using the numerical Jacobian matrix (Supplementary Table 3). BRET data are collected in three biological replicas and are given as mean ± SD.

**Reporting summary**. Further information on research design is available in the Nature Portfolio Reporting Summary linked to this article.

## Data availability

All data that support the findings of this study are available from the corresponding author upon reasonable request. The burst data from smFRET experiments and data from molecular dynamics simulations are available in Zenodo with the identifier https://doi.org/10.5281/zenodo.7722845[105].

## Code availability

All analyses of experimental smFRET data were performed in the software package PAM (PIE Analysis with MATLAB)[90]. The software is available as a source code, requiring MATLAB to run, or as pre-compiled standalone distributions for Windows or MacOS at http://www.cup.uni-muenchen.de/pc/lamb/software/pam.html and hosted in Git repositories under http://www.gitlab.com/PAM-PIE/PAM and http://www.gitlab.com/PAM-PIE/PAMcompiled. A detailed manual is located at http://pam.readthedocs.io. The version of PAM used for the analysis of smFRET data is also available in Zenodo with the identifier https://doi.org/10.5281/zenodo.7722845[105].

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

## Acknowledgements

A.L., A.B., and V.B. are thankful for the Ministry of Science and Higher Education of the Russian Federation (agreement #075-03-2023-106, project FSMG-2020-0003). IM acknowledges the UHasselt Special Research Fund. Measurements of surface expression and Gs-signaling were supported by the Russian Science Foundation (project no. 22-74-10036; https://rscf.ru/project/22-74-10036/). Computational simulations were supported by the National Natural Science Foundation of China, grant #32250410316 (to PO). We acknowledge the Advanced Optical Microscopy Centre at Hasselt University for support with microscopy experiments. Microscopy was made possible by the Research Foundation Flanders (FWO, projects G0B4915, G0B9922N, and G0H3716N).

## Author contributions

O.V. performed receptor expression. I.M. and O.V. performed receptor labeling, purification, and nanodisc reconstitution. T.G. and I.M. performed ensemble-TCSPC measurements. I.M. and Q.C. measured fluorescence spectra. Q.C. and J.He. prepared the instrumentation for smFRET experiments. I.M. performed smFRET experiments, analyzed smFRET data and drafted the manuscript. J.He., A.Ba., S.W., A.Bo., and V.B. contributed to the analysis of smFRET data. A.Ge. performed cell functional assay. PKh performed TSA functional assay with the contribution of A.M., A.L., A.Gu., and P.Ku. P.O. performed molecular dynamics simulations. I.M., V.B., J.He., V.C., T.G., A.Bo., A.Gu., A.M., P.Kh., and V.G. discussed the data, analysis, and contributed to writing the manuscript. I.M., V.B., J.He., V.C., T.G., and J.Ho. conceived the study. All authors commented on and edited the manuscript. V.B. and J.He. supervised the project.

## Competing interests

The authors declare no competing interests.
