## [Peer Review File · Communications Biology]

Reviewers' comments:

Reviewer #1 (Remarks to the Author):

This manuscript uses single molecule FRET to explore the millisecond timescale motion of the GPCR Adenosine 2A receptor in the ligand-free, agonist, partial agonist and antagonist bound states. The results show under all conditions three states fit (high efficiency (HF), mid-efficient (MF), low efficient (LF) FRET) best to the data, but the relative populations are ligand dependent, for example, HF is most populated for full agonist whereas MF for apo or antagonist bound. They conclude the LF is ligand resistant and may represent misfold.

I find this manuscript technically well done and very detailed. I think it is of significant interest to the GPCR field, especially the importance of ligand-induced dynamics. The authors use two dyes (donor/acceptor) and nanodisc incorporated. The method uses a diffusing system (so the nanodisc-incorporated receptor is not bound to the slide) which enables the millisecond timescale to be explored, specifically the movement between the MF and HF fits to a timescale of 400 ns. The limitation is, timescales greater than a few milliseconds cannot be resolved. But in many respects this manuscript complements or adds to the long timescale (seconds) work of Wei et al (ref 32). The authors state they do not add longer timescales to their work so as not to overfit, but I ask is the motion they see, consistent with the conformational changes that Wei et al see on the seconds timescale. Are they connected/correlated? How does the MD compare between the two studies? Or does the LF population add to the analysis at all? In short I am asking how do we reconcile the millisecond and second motions seen by smFRET.

In reading I did not find many errors. But please check references as for example ref 19 has its authors incomplete, and I only checked up on a few! I think the depth of the wells in the energy diagram in Figure 3B do not reflect the populations of the histogram. For example LF varies from 10 to 20% and is the same for all in the energy diagram and is substantial; the depth of 80% for HF of agonist is not proportionally consistent and so on.

I think Figure S10 and S11 have incomplete legends and/or are not right. For example S10 is two states and has curves suggesting three states (orange, red, green); the legend does not indicate orange but suggests a dark cyan.

Figure S11 appears correct and relates to Figure 3A. The legend of Figure 3A says this is 1 ms bins, but the analyses of 3A differ to the 1 ms bins in Figure S11. Why the difference?

Reviewer #2 (Remarks to the Author):

The work of Maslov et al. applies state-of-the-art single molecule FRET analysis techniques to hunt down sub-millisecond dynamics in four data sets recorded with FRET-labeled A2A adenosine receptors (A2AAR) in presence or absence of antagonistic or agonistic ligands. The design of the experiment is elegant. The A2AAR molecules are placed in lipid nanodiscs, freely diffusing in and out of the laser spot of a confocal setup. The authors see a clear change of transfer efficiency histograms upon binding of partial agonist LUF5834 and a slightly stronger change upon binding of the full agonist NECA. The further analysis of the data is challenging.

It turns out that all four recorded transfer efficiency histograms suffer from strong heterogeneous broadening. The reported fluorescence anisotropy decays reveal that the dye labels are not able to rotate freely. Hence, the FRET measurements do not directly report on inter-dye distances. This hampers to quite an extent the interpretation of the histograms in terms of conformational changes. In fact, the authors reveal in the discussion that transfer-efficiencies do change upon binding of agonists even in the opposite direction as expected based on available crystal structures of A2AAR.

The authors undertook MD simulations in order to find an explanation for this stark discrepancy. The simulations seem indeed to give some qualitative indication, however, a quantitative understanding of the histograms based on the simulations is not presented.

In view of this lack of understanding, I get the impression that the author's interpretation of their dynamics study, the main focus of the paper, is not standing on solid ground. The authors apply four methods to identify dynamics in the data: E vs donor lifetime, FRET-2CDE and BVA dynamic scores, and fFCS. They see significant dynamics for all methods. I find only the fFCS analysis convincing. The measured E-vs-lifetime distributions are very close to the static line. Similarly, the FRET-2CDE scores are close to ten, i.e., the value expected for static systems. The authors argue that the measured distributions deviate from ideal static distributions, obtained from simulations, significantly. However, how robust are these methods against possible systematic errors? The BVA score distributions are a bit more convincing, but only in the fFCS data dynamics is clearly visible. The authors show that the faster dynamics, in the lower microsecond range, can be attributed to photophysical processes, most likely blinking. I agree. The slower dynamics (~ 400 us) is observed only in presence of an agonist. The authors attribute this to conformational dynamics of A2AAR. This is the major finding of the paper.

Unfortunately, given the problems of interpreting the histograms described above, I find this interpretation rather speculative. The strong heterogeneous broadening, the long-lived fluorescence anisotropies of the dyes, and the unexpected direction of the transfer efficiency change upon binding of agonist are for me clear indications for massive problems with dye-lipid or dye-protein interactions. In this case, sticking dynamics are to be expected. Judging from the anisotropy decays, such dynamics seem to exist under all four conditions measured, but possibly it becomes only (weakly) detectable when the agonist is bound. Either because some of the dynamics shifts upon binding in to the accessible time window between blinking and diffusion dynamics, or the resulting FRET fluctuations become only strong enough for detection under these conditions. Still the observed effect is very weak. This or a similar scenario does not seem unlikely to me in view of the information given. In short, I am not convinced, that the 400-us-dynamics are due to conformation changes, that might be biologically relevant, or due to artifacts of the method.

The following PDA analysis assumes a well-defined Forster radius (with κ^2 averaged to $2/3$) and describes the heterogeneous broadening in terms of normal distributed inter-dye distances. Given the problems with the hindered rotation of the dyes, this procedure does not seem to give much physical insight. From this point of view, the proposed three-state model seems to be highly speculative. Independent evidence would be needed.

In summary, the main conclusions of the manuscript are in my opinion not well supported by the data. The main problem is the FRET-construct with its unfavorable dye dynamics. I understand that it might be extremely difficult to come up with a better construct. The presented analysis of the data is thorough and state of the art. However, the authors would need to present their conclusions in a much more conservative way. In its current form I cannot recommend publication of the manuscript in Nature Communications Biology.

Point-by-point answers.

Editor comments:

E1. Improve the connection between MD simulations and experiments with rigorous quantitative analysis as mentioned by both reviewers.

As the reviewers requested, we added a quantitative analysis of inter-dye distances observed in the MD simulations (Fig. S15). Our MD simulations capture only sub-microsecond dynamics of the dyes, while quantitative comparison with the experiment would require 1000 times longer simulations. In our replies and in the revised manuscript, we explained this limitation and the main qualitative conclusions that we draw from the MD simulations (lines 400-403, 415-418).

E2. Please follow the reliability and reproducibility checklist for MD simulations attached and ensure that the proper convergence and statistical analyses are provided. Please upload a completed checklist with a revised manuscript.

We have completed the checklist addressing the issues related to reliability and reproducibility. Particularly, we have carried out additional replications for each investigated system, updated the Fig. S14, conducted the convergence analysis by estimating the volume accessible for the fluorescent labels during the simulations (new Fig. S16), and provided additional methodological details aiming for better reproducibility (Supplementary Information, Molecular dynamics simulations section, lines 270-280, 305-309). Also, the trajectories of production simulations are now available at GitHub (https://github.com/porekhov/A2a_smFRET).

E3. Please resolve the technical issues with unfavorable dyes and interpretation as mentioned by Reviewer 2.

In our replies, we explained why there are no unfavorable dynamics of the dyes that compromise our data. To further confirm this, we provided a new analysis of burst-wise fluorescence anisotropy, which did not show the potentially dangerous sticking dynamics (new Fig. S17). We provided sound alternative explanations for indirect indications of dynamics of the dyes mentioned by the reviewer (see our answer R2-4 to the second reviewer).

In addition, as the reviewer requested, we performed extensive screening of the labeling positions (see our answer R2-6 to the second reviewer). In short, we tested 11 protein constructs and tried all sorts of possible label positions - after all these attempts, the FRET-construct (L225C^{9,27}/Q310C^{8,65}) originally selected in our study is the best.

Finally, we reflected the limitations of the study identified by the reviewer in the revised Discussion section (lines 472-488). In particular, we revised the text to warn readers that the FRET changes observed in our study should not be interpreted as changes in distance.

E4. If you decide to submit a revised version, we ask that you ensure your manuscript complies with our editorial policies. Please see our revision checklist for guidance on formatting the manuscript and complying with our policies. A comprehensive guide to our formatting requirements for final submissions is also available for your reference here.

We have updated the manuscript according to the provided guidance.

Reviewer 1 comments:

We are grateful to the reviewer for their comments and for spotting several inaccuracies in our manuscript. The comparison with recent A_{2A}AR studies suggested by the reviewer puts our work in the relevant context and complements our observations of agonist-induced increase of the receptor conformational dynamics.

R1-1. In many respects this manuscript complements or adds to the long timescale (seconds) work of Wei et al (ref 32). The authors state they do not add longer timescales to their work so as not to overfit, but I ask is the motion they see, consistent with the conformational changes that Wei et al see on the seconds timescale. Are they connected/correlated? How does the MD compare between the two studies? Or does the LF population add to the analysis at all? In short I am asking how do we reconcile the millisecond and second motions seen by smFRET.

Yes, indeed the work of Wei et al (ref 32) complements our study extending our observations to longer time scales. Both studies imply a 3-state model of receptor activation and embedded structural features of these states are consistent. In both cases protein activation results in the shift of the intracellular tip of the transmembrane helix TM6 with opening of the G-protein-binding cavity. The consistency of these studies goes beyond the structural aspects. Thus, in both cases apo-/antagonist-bound proteins contain detectable fraction of active-like state (called “State 3” in 32 and “high-FRET (HF) state” in our work). Similarly, both studies detect the increased conformational dynamics in the agonist-bound receptor. Finally, we observe a low-FRET (LF) state that appears to be locked in a long-lived state in the μ s-to-ms time scale of our study. This state probably corresponds to State 1 in 32 where its transitions to State 2 were detected in >100-ms time scale.

We have added the corresponding discussion to the manuscript (lines 392-395):

“Previous studies based on NMR^{19–21} and single-molecule fluorescence microscopy³² provide complementary insights into the dynamics of long-lived (>2 ms) A_{2A}AR conformations. Similarly to our study, both NMR²⁰ and single-molecule fluorescence microscopy³² detect agonist-induced increase of the receptor conformational dynamics”

Regarding the comparison of MD simulations in both studies, the employed MD protocols are similar in many details, including system setup, equilibration, and production simulations. In our work, we opted for the CHARMM36m forcefield and the CHARMM-GUI web-server (membrane embedding, equilibration protocol), both of which are state-of-the-art choices for the simulations of membrane proteins currently. Like Wei et al., we conducted our production simulations in triplicate, however, the simulation time of each replica was twice longer (1 μ s vs. 0.5 μ s in Wei et al.).

The general conclusions drawn from MD simulations also match in both studies regardless of different labeling positions used in Wei et al. and in our study. Whether the fluorescent label is attached to position 289 as in Wei et al. or to position 225 as in the present study, the label explores the cavity formed by the intracellular ends of several transmembrane helices (TMs) 2, 3, 6, and 7 in the active (agonist-bound) conformation of the protein (i.e., the G protein binding site). It is also a common place in both studies that the label adapts significantly

different orientation in the inactive conformation being mainly surrounded by solvent and pointed toward the bilayer.

R1-2. In reading I did not find many errors. But please check references as for example ref 19 has its authors incomplete, and I only checked up on a few!

We thank the reviewer and have proof-checked the reference list.

R1-3. I think the depth of the wells in the energy diagram in Figure 3B do not reflect the populations of the histogram. For example LF varies from 10 to 20% and is the same for all in the energy diagram and is substantial; the depth of 80% for HF of agonist is not proportionally consistent and so on.

In our experiments we observed transitions between various FRET states (LF, MF, and HF) within a given protein complex (apo, agonist-/antagonist-bound). These measurements allow us to calculate occupancies of the FRET states for a fixed complex and respective energy differences within each energy landscape. The energy levels between complexes (e.g. LF for apo vs LF for agonist-bound receptor) cannot be compared based on our data, since we did not experimentally observe transitions between apo and ligand-bound complex states.

To give a little bit more information: our energy diagram is sketched on the basis of FRET states populations (p_{LF} , p_{MF} , p_{HF}) determined for fFCS-constrained PDA model. From these values we can calculate the free energy differences between the minima:

$$E_{MF} - E_{LF} = -kT \ln \frac{p_{MF}}{p_{LF}}$$

$$E_{HF} - E_{LF} = -kT \ln \frac{p_{HF}}{p_{LF}}$$

From these equations relative depths for MF and HF can be determined for a given protein complex relative to E_{LF} , while E_{LF} cannot be determined in the experiments and may be different for different protein complexes.

Antagonist ZM241385			
p_{LF}	0.11	$\frac{E_{LF}}{kT}$	N/A
p_{MF}	0.74	$\frac{E_{MF} - E_{LF}}{kT}$	-1.9
p_{HF}	0.15	$\frac{E_{HF} - E_{LF}}{kT}$	-0.3
Apo			
p_{LF}	0.15	$\frac{E_{LF}}{kT}$	N/A
p_{MF}	0.64	$\frac{E_{MF} - E_{LF}}{kT}$	-1.5
p_{HF}	0.20	$\frac{E_{HF} - E_{LF}}{kT}$	-0.3

Partial agonist LUF5834			
p_{LF}	0.20	$\frac{E_{LF}}{kT}$	N/A
p_{MF}	0.25	$\frac{E_{MF} - E_{LF}}{kT}$	-0.2
p_{HF}	0.55	$\frac{E_{HF} - E_{LF}}{kT}$	-1.0
Full agonist NECA			
p_{LF}	0.12	$\frac{E_{LF}}{kT}$	N/A
p_{MF}	0.11	$\frac{E_{MF} - E_{LF}}{kT}$	0.1
p_{HF}	0.77	$\frac{E_{HF} - E_{LF}}{kT}$	-1.9

These results can be plotted with $E - E_{LF}$ as Y-coordinate:

Fig. 3B was sketched from this plot with slight modifications to improve its visual appearance without changing the main trends shown in the data.

To exclude misinterpretation of Fig. 3B noted by the reviewer we clearly state that the landscapes of relative energy are plotted with low-FRET as a reference state (lines 945-946):

“The landscapes of relative energy are drawn with low-FRET as a reference state”

R1-4. I think Figure S10 and S11 have incomplete legends and/or are not right. For example S10 is two states and has curves suggesting three states (orange, red, green); the legend does not indicate orange but suggests a dark cyan.

We edited the initially misleading figure captions. Initially, we referred to the same color as either “dark cyan” or “dark green” - in the revised manuscript we consistently use “dark cyan”. The orange line was initially introduced in the pre-last sentence of Fig. S10 caption - in the revised manuscript we moved this sentence closer to the beginning of the text.

Revised Fig. captions (Supplementary information, lines 410-419, 420-427, 428-439):

Fig. S10 PDA histograms for the model with two dynamic states. PDA histograms were fit with a sum of two states, allowing the low-FRET and high-FRET states to interconvert. The resulting fit (black line) is a sum of distributions simulated for molecules that stay in the low-FRET (dark cyan line), or high-FRET (red line) state during the entire simulated time-bin, and the distribution for molecules that sample both low-FRET and high-FRET states within the time-bin (orange line). The exchange time was set as a free fit parameter. Columns show PDA distributions for three different time-bin lengths (T), rows correspond to $A_{2A}AR$ in different ligand-bound or apo conditions. The experimental distributions are shown as grey bars. The fitting residuals are given on the top of each panel. For the global fit, $\chi^2_{red} = 10.3$.

Fig. S11 PDA histograms for the model with three static states. PDA histograms were fit with a sum of three non-interconverting (‘static’) states. The fitting curve (black line) is shown on top of the experimental distributions (grey bars). Simulated distributions for individual states are shown in light green (low-FRET), dark cyan (medium-FRET), and red (high-FRET) lines. Columns show PDA distributions for three different time-bin lengths (T); rows correspond to $A_{2A}AR$ in different ligand-bound or apo conditions. The fitting residuals are given on the top of each panel. For the global fit, $\chi^2_{red} = 3.2$.

Fig. S12 PDA histograms for the fFCS-constrained PDA model. PDA histograms were fit with a sum of three states, allowing the medium-FRET and high-FRET states to interconvert. The longer exchange time derived from fFCS ($\tau_2 = 390 \pm 80 \mu s$) was used for LUF5834 and NECA. The infinitesimally long exchange times (~ 100 ms) were used for ZM241385 and apo condition. The resulting fit (black line) is a sum of distributions simulated for molecules that stay in the low-FRET (light green line), medium-FRET (dark cyan line), or high-FRET (red line) state during the entire simulated time-bin, and the distribution for molecules that sample both medium-FRET and high-FRET states within a time-bin (orange line). Columns show PDA distributions for three different time-bin lengths (T), rows correspond to $A_{2A}AR$ in different ligand-bound or apo-conditions. The experimental distributions are shown as grey bars. The fitting residuals are given on the top of each panel. For the global fit, $\chi^2_{red} = 3.6$.

R1-5. Figure S11 appears correct and relates to Figure 3A. The legend of Figure 3A says this is 1 ms bins, but the analyses of 3A differ to the 1 ms bins in Figure S11. Why the difference?

We hope that after we revised the captions for Fig. S10, S11, and S12, it is clear that Fig. 3A relates to Fig. S12 - both represent fFCS-constrained dynamic PDA model. Notably, Fig. S11 shows a static three-state PDA model.

Fig. 3A shows the same data and analysis as the 1 ms bins in Fig. S12.

Reviewer 2 comments:

We thank the reviewer for the detailed and constructive revision of our work. The remarks about possible dye dynamics prompted a number of important refinements of the manuscript and ultimately led to a discussion about the possible limitations of FRET-studies for GPCR research. We are sure, it is a very important and timely discussion, not only for our work, but for the community at large.

R2-1. It turns out that all four recorded transfer efficiency histograms suffer from strong heterogeneous broadening. The reported fluorescence anisotropy decays reveal that the dye labels are not able to rotate freely. Hence, the FRET measurements do not directly report on inter-dye distances. This hampers to quite an extent the interpretation of the histograms in terms of conformational changes.

Although we agree with reviewer's statements, these are not arguments against our fundamental and solid interpretation that ligand-induced changes in FRET histograms report on conformational changes of the receptor. Indeed, both inter-dye distance and relative orientation of the dyes affect FRET efficiency and both change upon conformational changes of the receptor.

Notably, we can investigate structural dynamics of A_{2A}AR without a need for accurate distance determination. Many smFRET studies do not report absolute distances (e.g. Harris et al. 2022 PMID: 35194038; Liauw et al. 2022 PMID: 35775730; Schmid & Hugel 2020 PMID: 32697684; Habrian et al. 2019 PMID: 31804469; Stella et al. 2018 PMID: 30503205) and some single-molecule fluorescence-based methods, such as smPIFE, use other primary structural readouts rather than distances (Stennett et al. 2015 PMID: 26263254), yet all these studies provide valuable insights into structural dynamics of biomolecules.

In the original manuscript, we wrongly stated that absolute distances could be measured in our case. We now corrected this error. In the revised manuscript we explicitly state that our FRET changes should not be interpreted as distance changes (see new Discussion chapter "Limitations of the study", lines 475-481):

"Our nanosecond-time fluorescence depolarization measurements (Fig. S13, Table S7) and microsecond-time MD simulations (Fig. S14, Fig. S15) indicate that the reorientation of the dyes attached to A_{2A}AR upon a conformational change of the protein strongly affects the measured FRET efficiency. This means that changes in FRET efficiency should not be interpreted exclusively as a distance changes, and, particularly, apparent distances measured in PDA should only be considered as parameters of the fit, not as physical distances between the dyes."

We also revised PDA section to highlight that fitting parameters in the PDA are 'apparent distances' and should not be interpreted as physical distances between the dyes (see our reply to the query R2-5).

R2-2. In fact, the authors reveal in the discussion that transfer-efficiencies do change upon binding of agonists even in the opposite direction as expected based on available crystal structures of A_{2A}AR. The authors undertook MD simulations in order to find an explanation for this stark discrepancy. The simulations seem indeed to give some qualitative indication, however, a quantitative understanding of the histograms based on the simulations is not presented.

In the original manuscript, we used MD to show that FRET efficiencies observed in our experiment do not contradict crystal structures of the A_{2A}AR. Qualitatively, MD shows that preferred positions of the dyes within the accessible volume strongly influence FRET efficiencies. This conclusion is related to the fact that the lengths of the dyes' linkers are comparable to the change of distance between labeling positions in the crystal structures of agonist and antagonist-bound A_{2A}AR. Since the naive estimation based just on the distances between Ca atoms of the labeled residues ignores the relocation of the dyes, it cannot reliably predict the direction of FRET change. Our MD calculations show redistribution of labels upon protein activation and the inverse direction of FRET change observed in the experiment is no longer surprising.

For the revision, we histogrammed the inter-dye distances in the individual frames of our simulations (Fig. S15 shown below as Fig. R2-2-1). The histograms show that the inter-dye distance changes, approximately, from 5 nm to 3 nm upon receptor activation. Assuming R₀=49Å, we can calculate the corresponding FRET values (Fig. R2-2-2), which qualitatively supports the change from medium FRET to high FRET upon receptor activation. Thus, MD provides a plausible explanation for the observed increase in the FRET efficiency upon A_{2A}AR activation, as we claim in lines 415-418:

“Thus, our MD simulations provide a plausible explanation for the observed increase in the FRET efficiency upon A_{2A}AR activation and show that the observed FRET changes agree with the available crystal structures of A_{2A}AR”

We respectfully disagree with the reviewer that a quantitative comparison between MD and FRET is possible in our case, given the limitations present both in MD and FRET. The main limitation is that our MD is short compared to the time regime of the experiment (~1μs vs. ~1ms). It captures only fast dynamics of the dyes, but ignores slower dynamics of the dyes and the receptor. These slower dynamics will change the positions of the dyes and FRET values impeding the quantitative analysis. In addition, none of the forcefields commonly used in MD is optimized to represent the dynamics of the fluorescent dyes in the interface of protein, lipids and solution. Quantitative comparison with FRET is further complicated due to uncertainties in κ² and photophysics of the dyes in the protein/lipid environment that affect R₀.

To highlight that MD samples the dynamics of the dye on the different timescales than those observed in the experiment, we mention these timescales in the new “Limitations of the study” Discussion section (lines 475-478):

“Our nanosecond-time fluorescence depolarization measurements (Fig. S13, Table S7) and microsecond-time MD simulations (Fig. S14, Fig. S15) indicate that the reorientation of the dyes attached to A_{2A}AR upon a conformational change of the protein strongly affects the measured FRET efficiency”

Fig. R2-2-1. Inter-label distances calculated from MD simulations.

Inter-label distances calculated from triplicate MD simulations (I1-3 correspond to simulations of the inactive (based on PDB: 3PWH) protein; A1-3 – the fully active (targeted to PDB: 5G53) protein). In panel A, Atto647N is attached to L225C^{6,27}, Alexa488 is attached to Q310C^{8,65}; in panel B, the labels are swapped, i.e., Alexa488 is attached to L225C^{6,27}, Atto647N is attached to Q310C^{8,65}. Color code matches that in Figure S14, i.e., three active replicas are shown in orange, red, and ochre; three inactive replicas – in cyan, blue, and violet.

Fig R2-2-2. Distributions of FRET efficiencies calculated from the inter-label distances in MD simulations

FRET efficiency (E) calculated from triplicate MD simulations assuming $R_0=49 \text{ \AA}$. I1-3 correspond to simulations of the inactive (based on PDB: 3PWH) protein; A1-3 – the fully active (targeted to PDB: 5G53) protein. In panel A, Atto647N is attached to L225C^{6,27}, Alexa488 is attached to Q310C^{8,65}; in panel B, the labels are swapped, i.e., Alexa488 is attached to L225C^{6,27}, Atto647N is attached to Q310C^{8,65}. Color code matches that in Figure S14, i.e., three active replicas are shown in orange, red, and ochre; three inactive replicas – in cyan, blue, and violet.

R2-3. In view of this lack of understanding, I get the impression that the author's interpretation of their dynamics study, the main focus of the paper, is not standing on solid ground. The authors apply four methods to identify dynamics in the data: E vs donor lifetime, FRET-2CDE and BVA dynamic scores, and fFCS. They see significant dynamics for all methods. I find only the fFCS analysis convincing. The measured E -vs-lifetime distributions are very close to the static line. Similarly, the FRET-2CDE scores are close to ten, i.e., the value expected for static systems. The authors argue that the measured distributions deviate from ideal static distributions, obtained from simulations, significantly. However, how robust are these methods against possible systematic errors? The BVA score distributions are a bit more convincing, but only in the fFCS data dynamics is clearly visible. The authors show that the faster dynamics, in the lower microsecond range, can be attributed to photophysical processes, most likely blinking. I agree. The slower dynamics ($\sim 400 \text{ us}$) is observed only in presence of an agonist. The authors attribute this to conformational dynamics of A2AAR. This is the major finding of the paper.

We completely agree with the reviewer that only fFCS provides a solid ground for the dynamics. While BVA can also be convincing, the E vs donor lifetime can only provide some

hints about protein dynamics in our case. We wrote the original draft along the same vein, for instance, titling the subsections as:

“Fluorescence lifetime data suggest sub-millisecond conformational dynamics”, “FRET-2CDE and BVA confirm that agonists enhance conformational dynamics” and “fFCS reveals fast photophysics-related dynamics and slow agonist-induced dynamics in A_{2A}AR”.

To further emphasize this idea we have updated the paragraph in the Discussion section (lines: 336-353):

“Using various single-burst descriptors and time-resolved analysis methods for quantifying FRET dynamics, we revealed sub-millisecond conformational dynamics in A_{2A}AR. Slight deviations of bursts from the ‘static FRET line’ on the FRET efficiency versus donor fluorescence lifetime plot hinted at nanosecond-millisecond dynamics for the apo-A_{2A}AR and A_{2A}AR with each of the used ligands (Fig. 2B), although can equally be attributed to undefined systematic errors. FRET-2CDE analysis suggested more pronounced conformational dynamics in the agonist-bound A_{2A}AR than in the apo or antagonist-bound A_{2A}AR (Fig. 2D). BVA confirmed that the variations of FRET efficiency among ~100 μs time-bins exceed the level expected from shot-noise (Fig. 2F). Finally, fFCS clearly confirms the dynamics nature of the data and demonstrated two components in A_{2A}AR dynamics: fast microsecond-time (3 ± 20 μs) dynamics present in all samples and assigned mostly to dyes photophysics and slower (390 ± 80 μs) dynamics evoked by agonists (Fig. 2H). It is fFCS that puts all our findings in a single self-consistent picture: both fast and slow dynamics contribute to the deviation of bursts from the ‘static FRET line’, however the fast dynamics make limited contributions to the FRET-2CDE scores and to the BVA distribution deviations because of their 10-fold faster timescale compared to the temporal resolution of these techniques. Meanwhile, the slower dynamics evoked with the agonists explains the increased dynamics scores in FRET-2CDE and BVA for the agonist-bound A_{2A}AR.”

R2-4. Unfortunately, given the problems of interpreting the histograms described above, I find this interpretation rather speculative. The strong heterogeneous broadening, the long-lived fluorescence anisotropies of the dyes, and the unexpected direction of the transfer efficiency change upon binding of agonist are for me clear indications for massive problems with dye-lipid or dye-protein interactions. In this case, sticking dynamics are to be expected. Judging from the anisotropy decays, such dynamics seem to exist under all four conditions measured, but possibly it becomes only (weakly) detectable when the agonist is bound. Either because some of the dynamics shifts upon binding in to the accessible time window between blinking and diffusion dynamics, or the resulting FRET fluctuations become only strong enough for detection under these conditions. Still the observed effect is very weak. This or a similar scenario does not seem unlikely to me in view of the information given. In short, I am not convinced, that the 400-us-dynamics are due to conformation changes, that might be biologically relevant, or due to artifacts of the method.

The potential artifacts and limitations mentioned by the reviewer are inherent in FRET measurements and cannot be completely eliminated. For our case of a small transmembrane protein, which has only few residues available for labeling and undergoes relatively small structural changes compared to established systems these limitations are indeed more challenging. In the revised manuscript, we discuss the limitations of our

approach and alternative explanations for the dynamics observed in our data (see “Limitations of the study” chapter in Discussion, lines 472-488):

“In this study, we used single-molecule FRET to investigate the dynamics of the A_{2A}AR. The intrinsic limitation of FRET as a label-based method is that the dynamics of dyes and protein cannot be completely separated based on fluorescence data. Our nanosecond-time fluorescence depolarization measurements (Fig. S13, Table S7) and microsecond-time MD simulations (Fig. S14, Fig. S15) indicate that the reorientation of the dyes attached to A_{2A}AR upon a conformational change of the protein strongly affects the measured FRET efficiency. This means that changes in FRET efficiency should not be interpreted exclusively as a distance changes, and, particularly, apparent distances measured in PDA should only be considered as parameters of the fit, not as physical distances between the dyes. Additionally, we cannot completely exclude that the dynamics of the dyes contribute to the observed 390 μs dynamics. However, fluorescence depolarization measurements suggest that the orientational freedom of the dyes is almost ligand-independent (Fig. S13, Table S7) and burst-wise anisotropy measurements do not indicate multiple long-lived states of the dyes on the millisecond timescale (Fig. S17). Therefore, we assign the agonist-induced dynamics observed in our data to the dynamics of the receptor. This interpretation is supported by previous NMR-based studies that also observed agonist-induced dynamics in A_{2A}AR on a sub-millisecond timescale^{17,20}”

Even though the fundamental limitations exist we find our initial interpretation of FRET data more sound than the scenarios suggested by the reviewer.

First, heterogeneous broadening of FRET distributions besides dye-lipid/dye-protein dynamics can be naturally explained by conformational flexibility of the receptor, which was previously reported in the literature. GPCRs undergo substantial fluctuations even within a single conformational state as demonstrated by various experimental techniques (e.g. Wingler et al. 2019 PMID: 30639099; Preininger et al. 2013 PMID: 23602809; Zokher et al. 2012 PMID: 22748765; Manglik et al. 2015 PMID: 25981665) as well as MD (Latorraca et al. 2016 PMID: 27622975).

Second, incomplete anisotropy decay is also expected in our case since reorientation of the dyes is sterically limited in proximity of lipid bilayer and protein even if the dye does not interact with protein or lipids. Moreover, anisotropy depolarization monitors only very fast motions of the dyes (~ns) and can not reliably indicate 10⁵ times slower sticking dynamics on the timescales where we observe agonist-induced dynamics (~390 μs). To address the reviewer’s concern that limited reorientational freedom of the dyes can be a signature of slow (~ 390 μs) dynamics of dye-protein or dye-lipid interaction we performed burst-wise analysis of the fluorescence anisotropy in double-labeled molecules. If sticking/unsticking of the dyes were a problem we would expect that burst-wise anisotropies would reflect different long-lived states as multiple peaks in the anisotropy distributions. Contrarily, we do not see neither different anisotropy-states nor ligand-induced changes of the distribution (see new Fig. S17 given below as Fig. R2-4-1). Even though the sensitivity of this approach is limited this is the most direct way to test the reviewer’s hypothesis and we do not see any signature of a massive problems with dye-lipid/dye-protein interactions in our experiment, as the reviewer suggests.

Third, the direction of the observed FRET changes is naturally explained with our molecular dynamics simulations. As we discussed in R2-2, MD shows our observations in FRET experiment agree with the available crystal structures. Therefore, the observed FRET changes do not indicate any problem in the FRET experiment.

Finally, the biological relevance of 400-us-dynamics is confirmed by the cross-validation with NMR studies (ref 17 and ref. 20, listed in lines 367-369) that also suggest agonist-induced dynamics of $A_{2A}AR$ in the sub-millisecond timescale with a completely different experimental setup. These NMR-based studies demonstrate sub-millisecond agonist-induced dynamics qualitatively and on the ensemble level and we use single-molecule FRET to quantify the exchange time of these dynamics and build a quantitative mechanistic model of the receptor activation.

Figure R2-4-1. Distributions of the burst-wise fluorescence lifetime (τ) and anisotropy (r) for fluorophores in (A) donor-only and (B) acceptor-only $A_{2A}AR$ populations.

R2-5. The following PDA analysis assumes a well-defined Forster radius (with kappa-squared averaged to 2/3) and describes the heterogeneous broadening in terms of normal distributed inter-dye distances. Given the problems with the hindered rotation of the dyes, this procedure does not seem to give much physical insight. From this point of view, the proposed three-state model seems to be highly speculative. Independent evidence would be needed.

Since we do not interpret the apparent distances obtained from PDA and use them merely as fitting parameters the issues mentioned above do not challenge our conclusions. Indeed, PDA assumes that the ratio R/R_0 is normally distributed, but this, in general, does not imply neither constant R_0 , nor $\kappa^2 = 2/3$. Since the shape of R/R_0 distribution in the underlying states cannot be directly observed and to some extent is a matter of definition, the normal distribution is the best commonly accepted approximation.

We use PDA to quantify the FRET histograms and find the populations of the observed FRET states. Some qualitative biological conclusions are apparent from the distributions and would be similar if we use an alternative analysis technique (e.g. fitting of the distributions with sums of Gaussians): to describe all datasets we need three FRET states spanning over low-FRET, medium-FRET and high-FRET region, and agonists clearly increase the population of high-FRET state and decrease the population of medium-FRET - PDA allows us to make the same conclusion on quantitative basis.

The added value of PDA is that it allows us to analyze the dynamics of the receptor via global analysis of the histograms calculated with varying binning times. Even though we found out that in our case PDA is insensitive to the receptor's dynamics, this negative result is important for the understanding of our data. Moreover, we show that PDA histograms can also be fitted with a model consistent with our findings from fFCS. Since simple Gaussian fitting would not let us test the consistency between the observed FRET histograms and fFCS findings, we find PDA useful and insightful in our case.

To warn the reader that R and σ in PDA should not be directly interpreted as inter-dye distance and its standard deviations, but rather as fitting parameters that determine position and width of the FRET histogram, we highlighted this in a new paragraph "Limitations of the study" in Discussion (lines 475-481):

"Our nanosecond-time fluorescence depolarization measurements (Fig. S13, Table S7) and microsecond-time MD simulations (Fig. S14, Fig. S15) indicate that the reorientation of the dyes attached to A_{2A}AR upon a conformational change of the protein strongly affects the measured FRET efficiency. This means that changes in FRET efficiency should not be interpreted exclusively as a distance changes, and, particularly, apparent distances measured in PDA should only be considered as parameters of the fit, not as physical distances between the dyes;"

We now refer to the fitting parameters in PDA as 'apparent distances' in the paragraph given above and in the captions of Tables S4-S6;

We also removed R values from Fig. 3 caption to avoid the confusion (lines 935-938):

"The resulting fit (black line) is a sum of distributions simulated for molecules that stay in the LF (light green line), MF (dark cyan line), or HF (red line) state during the entire simulated time-bin, and the distribution for molecules that sample both MF and HF states within the time-bin (orange line)"

Finally, we revised lines 304-306:

"Using the fFCS-constrained model in PDA we determined the mean values and variances of the FRET efficiency and populations for each PDA state under the apo and ligand-bound conditions (Fig. 3A, Fig. S12, and Table S6)"

R2-6. In summary, the main conclusions of the manuscript are in my opinion not well supported by the data. The main problem is the FRET-construct with its unfavorable dye dynamics. I understand that it might be extremely difficult to come up with a better construct. The presented analysis of the data is thorough and state of the art. However, the authors would need to present their conclusions in a much more conservative way. In its current form I cannot recommend publication of the manuscript in Nature Communications Biology.

In our reply to query R2-4, we explained why the unfavorable dynamics of the dyes is unlikely to be present in our data. The most direct way to test it, the analysis of the burst-wise anisotropies, did not show different dye states in agonist-bound sample.

Generally, we agree that possible artifacts connected with dynamics of the dyes cannot be completely excluded in any FRET experiment and in our case they should be discussed in more detail. We presented our conclusions in a more conservative way by in-text changes and addition of “Limitation of the study” subchapter to discussion in the revised manuscript where we directly discussed possible concerns.

Limitations of the study (line 472-488):

In this study, we used single-molecule FRET to investigate the dynamics of the A_{2A}AR. The intrinsic limitation of FRET as a label-based method is that the dynamics of dyes and protein cannot be completely separated based on fluorescence data. Our nanosecond-time fluorescence depolarization measurements (Fig. S13, Table S7) and microsecond-time MD simulations (Fig. S14, Fig. S15) indicate that the reorientation of the dyes attached to A_{2A}AR upon the conformational change of the protein strongly affects the measured FRET efficiency. This means that changes in FRET efficiency should not be interpreted exclusively as a distance changes, and, particularly, apparent distances measured in PDA should only be considered as parameters of the fit, not as physical distances between the dyes. Additionally, we cannot completely exclude that the dynamics of the dyes contribute to the observed 390 μs dynamics. However, fluorescence depolarization measurements suggest that the orientational freedom of the dyes is almost ligand-independent (Fig. S13, Table S7) and burst-wise anisotropy measurements do not indicate multiple long-lived states of the dyes on the millisecond timescale (Fig. S17). Therefore, we assign the agonist-induced dynamics observed in our data to the dynamics of the receptor. This interpretation is supported by previous NMR-based studies that also observed agonist-induced dynamics in A_{2A}AR on a sub-millisecond timescale^{17,20}.

To test the reviewer's idea that alternative labeling construct may provide better results in terms of unfavorable dye dynamics we did a labeling position scan. As mentioned by the reviewer, the design of FRET-constructs for our system is indeed extremely challenging: A_{2A}AR is a small transmembrane protein with only few residues accessible for labeling, also A_{2A}AR undergoes only minor structural changes compared to other systems typically analyzed via smFRET. We have tested 11 protein constructs (see Table R2-6-1 below) and tried all sorts of possible label positions (various helices and different protein surfaces, see Fig. R2-6-1).

Some protein constructs result in unfolded protein, for others cysteines were not available for labeling resulting in low labeling efficiency. In the most successful cases, FRET efficiency was either too high or too low to sense conformational changes of the protein. After all attempts,

L225C^{6.27}/Q310C^{8.65} FRET-construct, originally selected in our study from theoretical considerations, is the best.

Consequently, the FRET-construct used in our study provides the best results possible for the smFRET with our research object. As the reviewer concluded, we provide a thorough and state-of-the-art analysis for this complicated FRET data. In the revised version we also discuss possible concerns related to the analysis. We are confident that our work provides a highly valuable example of what information about GPCRs can be obtained with the emerging smFRET technique and what limitations should be expected. Keeping in mind that GPCRs is the most pharmacologically important protein class, we are confident that our work is worth wide sharing within the scientific community despite concerns raised by the reviewer.

Mutation	Label protein yield, µg per 0.5L	Labeling efficiency, %	Agonist-induced FRET changes
E228C	<10		
V229C	50		
A289C	110	3	
R291C	240	7	
A205C	280	8	
L225C	180	25	selected for introduction of 2nd mutation
L225C/T41C	70		
L225C/R107C	280	0.4	
L225C/V8C	170	7	low FRET unrelated to ligand added
L225C/T117C	120	5	high FRET unrelated to ligand added
L225C/Q310C	170	5	

Table R2-6-1. Screening of labeling position for A_{2A}AR. Labeling efficiency for single cysteine mutants shows percentage of labeled molecules relative to all protein molecules, for double mutants - percentage of double labeled molecules relative to all labeled molecules.

Figure R2-6-1. Screening of labeling position for A_{2A}AR. Positions experimentally tested in the study are labeled red.

PS.

We also corrected a typo in Fig. S1(E):

Before:

After:

6

REVIEWERS' COMMENTS:

Reviewer #1 (Remarks to the Author):

The authors have addressed the issues I raised and explained the analyses that I was unclear on.

Reviewer #4 (Remarks to the Author):

Dr. Borshchevskiy and colleagues addressed my concerns and those of Reviewer #1 thoroughly and satisfactorily. The manuscript has been improved substantially. I highly appreciate that the authors added the section about possible limitations of the study. I recommend publication in Communications Biology.